# Think2SQL: Blueprinting Reward Density and Advantage Scaling for Effective Text-to-SQL Reasoning

**Simone Papicchio**                                    *simone.papicchio@polito.it*
*Politecnico di Torino, Italy & EURECOM, France*

**Simone Rossi**                                          *simone.rossi@eurecom.fr*
*EURECOM, France*

**Luca Cagliero**                                          *luca.cagliero@polito.it*
*Politecnico di Torino, Italy*

**Paolo Papotti**                                          *papotti@eurecom.fr*
*EURECOM, France*

**Reviewed on OpenReview:** https://openreview.net/forum?id=NxU1KWnpOG

## Abstract

While Large Language Models (LLMs) have advanced the state-of-the-art in Text-to-SQL, robust reasoning in complex, multi-table environments remains a bottleneck for parameter-efficient models. This paper presents a systematic empirical study on injecting reasoning capabilities into Text-to-SQL through the lens of Reinforcement Learning with Verifiable Rewards (RLVR) for the `Qwen3` model family. We uncover a critical **interplay between reward density, advantage scaling, and model capacity**. Our analysis yields four primary insights. First, we propose a novel *execution-guided dense reward* function that significantly outperforms binary signals and existing state-of-the-art rewards by providing granular feedback at the instance level. Second, we analyze the mechanics of advantage calculation, demonstrating that while large models thrive on sparse signals with aggressive advantage scaling, smaller models require dense rewards and conservative scaling to improve Text-to-SQL performance. Third, we evaluate the impact of cold start showing that distillation does not always benefit RLVR performance, and supervised fine-tuned models are prone to *distributional mimicry*. Fourth, we map the Pareto frontier of training efficiency, providing insights for optimizing Text-to-SQL reasoning under computational constraints. Our findings culminate in the `Think2SQL` family: our 4B-parameter model demonstrates reasoning capabilities competitive with state-of-the-art models such as `o3`. We release our models, datasets, and code to create a blueprint for RLVR optimization in Text-to-SQL at https://github.com/spapicchio/Think2SQL.

## 1 Introduction

The exponential growth of relational data has motivated the development of natural language interfaces that allow non-technical users to query complex databases (Floratou et al., 2024). This task, known as Text-to-SQL, involves translating natural language into executable SQL queries, a process that requires both deep semantic understanding and the ability to map first-order logic onto intricate multi-table schemas (Badaro et al., 2023; Fan et al., 2024). While Large Language Models (LLMs) have significantly advanced the state-of-the-art on benchmarks such as BIRD (Li et al., 2024b) and SPIDER (Yu et al., 2018), their performance remains highly sensitive to model scale and the specific training paradigms employed for task adaptation. Beyond academic interest, Text-to-SQL has become a critical enterprise capability: major cloud providers now offer production Text-to-SQL features (e.g., Google BigQuery, Oracle Select AI), and companies such as LinkedIn

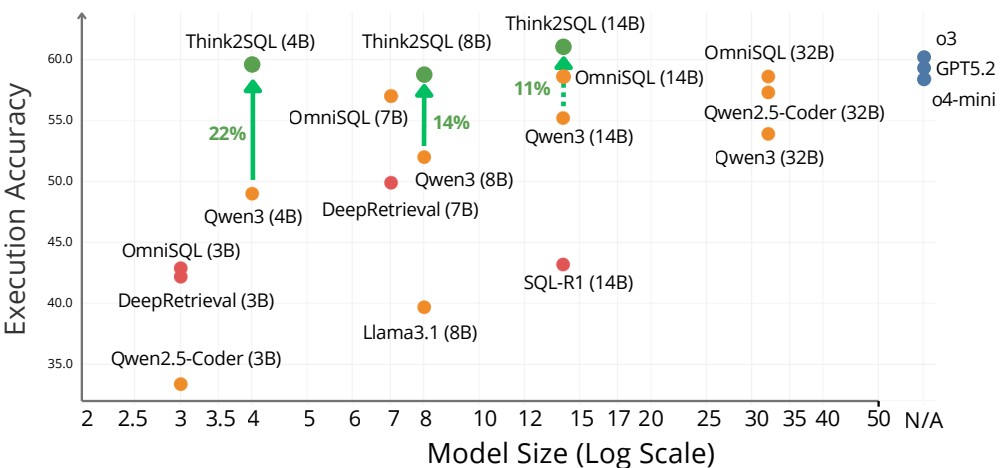

Figure 1: **Execution vs. model size (log scale) for various open- and closed- models on the BIRD-Dev set.** Blue points indicate models with general reasoning, orange points indicate models without reasoning, red points indicate task-specific reasoning, and green points indicate our models (`Think2SQL-4B/8B/14B`). Vertical green lines indicate the relative performance gains achieved by RLVR, trained on BIRD-TRAIN. All models are instructed versions. `OpenAI` models' sizes are not publicly known.

have deployed internal Text-to-SQL chatbots to democratize data access across their organizations (Chen et al., 2025a), underscoring the practical urgency of improving training methodologies for this task.

To improve performance, LLMs can be adapted to the Text-to-SQL task through various training techniques. Supervised Fine-Tuning (SFT) remains the dominant strategy (Wei et al., 2022; Li et al., 2024a), yet it is fundamentally constrained by the availability of high-quality, human-annotated datasets. For smaller LLMs, this reliance on static examples often leads to limited generalization; these models frequently struggle with logical inference on out-of-distribution queries, even after extensive fine-tuning (Papicchio et al., 2025b). Reinforcement Learning with Verifiable Rewards (RLVR) has recently emerged as a compelling alternative, treating SQL generation as a decision-making process where the model learns through iterative exploration and rule-based feedback (DeepSeek-AI et al., 2025). By interacting with a database engine during training, the model can potentially discover valid reasoning paths (not present in the SFT training set) that favor actions (e.g., SQL queries) that yield higher rewards.

Despite this potential, the application of RLVR to Text-to-SQL is currently hindered by a lack of standardized reward mechanisms. Existing approaches are often polarized between sparse binary signals based on execution accuracy (Ma et al., 2025) and heuristic-heavy composite rewards that rely on syntactic similarity or stochastic LLM-as-a-judge evaluations (Pourreza et al., 2025). These methods overlook a fundamental dimension of RLVR optimization: the interplay between reward density, advantage scaling, and model capacity. This oversight is particularly detrimental to smaller models, which often fail to converge with sparse rewards due to insufficient exploration guidance.

In this work, we address these limitations by providing a systematic empirical characterization of the RLVR landscape for the `Qwen3` model family representing the current state-of-the-art for open-source reasoning (Artificial Analysis, 2026). Through an extensive study involving approximately 20,000 GPU hours, we demonstrate that there is no universal "best" reward signal; rather, the optimal reward density is intrinsically tied to the model's parameter count. Our findings reveal that smaller models, such as `Think2SQL-4B`, benefit disproportionately from execution-guided dense feedback and conservative scaling, whereas larger models can generalize from sparse signals and more aggressive scaling. Furthermore, we extend this analysis to the *Pareto frontier of training efficiency*, characterizing the trade-offs between computational budget and final model quality. The proposed advantage scaling and reward shaping strategies are inherently model-agnostic and can be applied to other architectures. However, our evidence is limited to the `Qwen3` model family. A preliminary analysis on the Qwen2.5 family is present in Papicchio et al. (2025c).

To guide this investigation, we structure our analysis around three central research questions. **RQ1: Scale-Density Interdependence**. How do reward sparsity and advantage scaling interact with model capacity, and do smaller models necessitate denser signals for stable convergence? **RQ2: The cold-start problem.** How does the cold start affect policy convergence and the discovery of novel reasoning paths during RLVR? **RQ3: The Pareto Frontier.** What is the optimal trade-off between training duration and model quality across different RLVR configurations? Our contributions are summarized as follows: (i) We propose a lightweight execution-guided dense reward function for Text-to-SQL, which surpasses both binary signals and existing state-of-the-art rewards by delivering granular, instance-level feedback. (ii) We conduct a comprehensive analysis of the interplay between reward sparsity, model scale, and reward scaling strategies, revealing that the optimal reward signal is model-dependent rather than universal. (iii) We introduce `Think2SQL`, a reasoning-augmented model family that enables a 4B parameter model to rival proprietary frontier models such as `o3` (Fig. 1). (iv) We release a novel SFT dataset distilled from `Gemini3-Flash` and evaluate its influence on RLVR performance. (v) We map the Pareto frontier of training efficiency, offering actionable insights for optimizing Text-to-SQL reasoning under constrained computational resources.

The rest of the paper is organized as follows. Section 2 reviews the state-of-the-art works. Section 3 describes the proposed Text-to-SQL approach. Section 4 summarizes the main experimental results. Sections 5 and 6 draw conclusions and discuss the main limitations of the present work, respectively.

## 2 Related Work

**Text-to-SQL Pipelines.** Modern Text-to-SQL frameworks generally adopt a tripartite architecture (Shi et al., 2025; Chung et al., 2025): (i) *pre-processing* via schema augmentation (Li et al., 2025) and linking (Talaei et al., 2024); (ii) *inference* using multi-step reasoning (Pourreza & Rafiei, 2023) or modular decomposition (Wang et al., 2025a); and (iii) *post-processing* through self-correction (Pan et al., 2023). While effective, these modular pipelines often rely on frozen LLMs, missing the optimization benefits of direct preference tuning or reinforcement learning.

**RLVR in Text-to-SQL.** Reinforcement Learning with Verifiable Rewards (RLVR) has emerged as a high-performance paradigm for aligning SQL generation (Ma et al., 2025; Yao et al., 2025). Current approaches vary significantly in their training regimes and reward designs. For instance, SQL-R1 (Ma et al., 2025) employs a two-stage SFT-then-RL strategy with a complex composite reward including output-length penalties. Conversely, Arctic-SQL (Yao et al., 2025) and Reasoning-SQL (Pourreza et al., 2025) utilize single-stage RL, but differ in reward granularity: Arctic-SQL uses a near-binary signal based on execution and syntax, while Reasoning-SQL integrates diverse heuristics like N-gram similarity and LLM-as-a-judge.

**The Sparsity Gap and Scaling Interplay.** Despite these advances, the field remains fragmented, with reward signals polarized between sparse binary indicators (Yao et al., 2025) and heuristic-heavy composites (Pourreza et al., 2025). Crucially, the relationship between reward density and model capacity remains under-explored. Our work bridges this gap through an extensive empirical analysis using the `Qwen3` model family, uncovering a fundamental interplay between reward density, model scale, and scaling strategies. By characterizing how denser signals benefit different parameter counts, we provide a systematic blueprint for optimizing RLVR trajectories in Text-to-SQL tasks.

**RLVR in other domains.** The efficacy of RLVR has been validated across diverse modalities, mirroring our findings while highlighting Text-to-SQL's unique requirements. Recent studies in Vision-Language-Action (VLA) and VLMs confirm that RL generalizes better than SFT by exploring latent capabilities (Liu et al., 2025b; Lu et al., 2026; Chen et al., 2025b). While research on R1-like models shows that SFT provides an immediate boost (Zhang et al., 2026), other work demonstrates that it can create an "SFT quagmire" in which misleadingly high scores constrain the RL exploration space (Kang et al., 2025). Our work confirms this ceiling effect (RQ2), showing that base models often surpass SFT-initialized ones in peak performance. Furthermore, while general RL reasoning may be driven by high-entropy tokens (Wang et al., 2025b), we show that in Text-to-SQL, this is scale-dependent: smaller models require dense, execution-guided feedback to prevent training stalls, whereas larger models thrive on sparse signals and aggressive scaling. In addition, recent work has explored the connection between outcome-based and process-based reward signals in RL for

reasoning (Feng et al., 2025). Our work is complementary, as we focus on the outcome-based reward design space, but our findings that dense rewards help small models but might hurt large ones may inform PRM research as well.

## 3 Methodology

This section is organized as follows: § 3.1 formalizes the Text2SQL problem and the related notation; § 3.2 details the SFT procedure and describes the training dataset; finally, § 3.3 and § 3.4 thoroughly describe the RLVR training framework and the reward functions adopted in our work, respectively.

### 3.1 Problem statement

Text-to-SQL translates a natural language (NL) question $\boldsymbol{x}$ into an SQL query $\boldsymbol{y}$ based on a database schema $\mathcal{S}$, which defines tables, attributes, and data types. Auxiliary context $\mathcal{M}$, such as metadata or prior query examples, may also be included.

Let $\pi_{\boldsymbol{\theta}}$ denote a Large Language Model (LLM) parameterized by $\boldsymbol{\theta}$, formalized as a probabilistic autoregressive decoder-only model (Radford et al., 2018). The model $\pi_{\boldsymbol{\theta}}$ defines a conditional distribution over the SQL query $\boldsymbol{y} = (y_1, \ldots, y_T)$ given the input $\boldsymbol{x}$ and the schema context:

$$\pi_{\boldsymbol{\theta}}(\boldsymbol{y} \mid \boldsymbol{x}, \mathcal{S}, \mathcal{M}) = \prod_{t=1}^{T} \pi_{\boldsymbol{\theta}}(y_t \mid \boldsymbol{x} \| \mathcal{S} \| \mathcal{M} \| \boldsymbol{y}_{<t}), \tag{1}$$

where $\|$ denotes the concatenation operator. Following prior works (Li et al., 2025; Yao et al., 2025; Ma et al., 2025), we enrich the schema representation with descriptions and attribute examples to enhance model comprehension.

Standard Text-to-SQL evaluation using **Execution Accuracy (EX)** (Li et al., 2024b) often encounters semantic inconsistencies due to its reliance on set-based comparisons. Specifically, existing EX implementations exhibit two major shortcomings: (i) *Multiplicity Loss*, where queries with and without `DISTINCT` are incorrectly treated as equivalent, and (ii) *Column Sensitivity*, where valid permutations of the projected columns are unfairly penalized. Consider the resulting tables of the executed ground-truth and predicted queries (represented as lists of tuples), the set-based comparison applies only to the outer collection, leaving the metric vulnerable to the aforementioned issues. To address these limitations, we propose a **Refined EX** metric that incorporates bag semantics, preserving row multiplicity, and ensures invariance to attribute ordering by sorting tuples prior to comparison. This approach guarantees that the evaluation focuses on the informational content of queries rather than their structural formatting. To support reproducible and standardized benchmarking, we introduce NL2SQLEVAL[1], an extensible Python toolkit for modular SQL evaluation. All references to *EX* in this work pertain to our refined evaluation procedure. Additional technical details are provided in § A.

### 3.2 Supervised Fine-Tuning

SFT aims to train the model to maximize the likelihood associated with next-token prediction and defined in Eq. (1). This sequence consists of the system prompt, user prompt, and distilled reasoning trace with the corresponding answer.

The loss function is computed exclusively on the distilled reasoning traces and answers, neglecting the input tokens, to optimize training efficiency (Chiang et al., 2023; Yu et al., 2024). This is particularly beneficial as the distilled traces are significantly longer than the input sequence (Shi et al., 2024). Additional details on the SFT objective can be found in § B.1.

**Training dataset $\mathcal{D}_{SFT}$.** We construct our training set, hereafter called $\mathcal{D}_{SFT}$, by distilling reasoning traces from `Gemini3-Flash` (Google, 2026a) using the BIRD benchmark (Li et al., 2024b) as a foundation.

---

[1]https://github.com/spapicchio/NL2SQLEvaluator

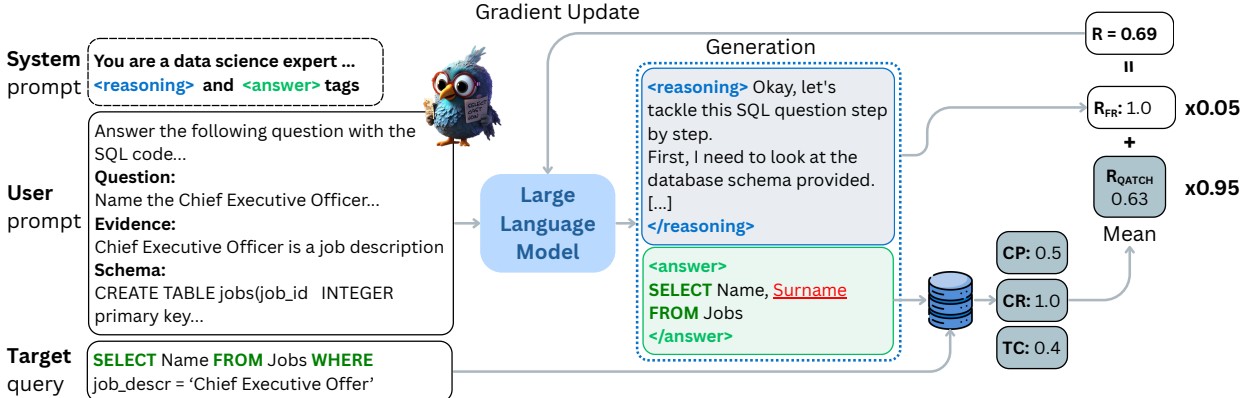

Figure 2: **Overview of the RLVR algorithm.** An LLM, either a base or SFT model, generates multiple outputs per step, each comprising a reasoning trace and a predicted SQL query. Rewards are computed as a weighted sum of format adherence and SQL correctness, incorporating metrics such as *Cell Precision* (CP), *Cell Recall* (CR), and *Tuple Cardinality* (TC) (2023). These rewards guide model updates via DAPO.

This benchmark provides $9,428$ examples across 69 databases and 37 domains; each entry pairs a natural language (*NL*) question and *evidence* with a ground-truth *SQL* query. After the removal of erroneous SQL for quality pre-processing (Muennighoff et al., 2025), we apply the OmniSQL (Li et al., 2025) prompt template to extract the model's output. The model is configured with a medium thinking level and a temperature $\tau = 1.0$. Given that `Gemini3-Flash` provides a summarized abstraction of its internal reasoning rather than a raw chain-of-thought, we preserve both the distilled reasoning summary and the generated SQL statement. This process yields $5,682$ high-quality, reasoning-aligned examples. We provide a representative sample in § B.1 and host the complete dataset[2] under the Gemini API license (Google, 2026b).

### 3.3 Reinforcement Learning with Verifiable Reward (RLVR)

Reinforcement Learning with Verifiable Reward (RLVR) (Lambert et al., 2024; DeepSeek-AI et al., 2025; Team et al., 2025) extends the RLHF (Ouyang et al., 2022) framework by replacing the learned reward model with rule-based, verifiable rewards. This paradigm reduces reward hacking and spurious correlations (Gao et al., 2023; Everitt et al., 2021) and has been shown to enhance specific cognitive behaviors (Gandhi et al., 2025). Despite the challenge of reward sparsity, where feedback is only provided upon strict constraint satisfaction, RLVR is highly effective for tasks with verifiable objectives, such as code or SQL generation (Le et al., 2022; Gehring et al., 2024; Chen et al., 2023).

As shown in Fig. 2, the model receives a prompt $x$ consisting of a *question*, *evidence*, and *database schema*, and samples a group of $G$ outputs $\{y_1, y_2, \ldots, y_G\}$. Each generation is assigned a reward, $R_i$, based on the execution accuracy of the generated SQL and on format adherence. To optimize the policy, we compute the relative advantage $A_i$ using one of three normalization strategies: (i) **Group Scaling**, where rewards are normalized within each prompt group, $A_i = \frac{R_i - \text{mean}(\{R\}_G)}{\text{std}(\{R\}_G)}$, following the GRPO approach (DeepSeek-AI et al., 2025); (ii) **Batch Scaling**, where rewards are normalized across the entire training batch to enable a more robust reward shaping (Liu et al., 2025d); and (iii) **No Scaling**, where there is no variance normalization to mitigate question-level difficulty bias (Liu et al., 2025c). Further details about optimization policies and the training prompt are given in § B.2 and § C, respectively.

### 3.4 Reward Design

In Reinforcement Learning from Verifiable Rewards (RLVR), the design of reward signals is critical to avoid optimization collapse and reward hacking (Everitt et al., 2021). In Text-to-SQL tasks, Execution Accuracy is

---

[2]https://github.com/spapicchio/Think2SQL

the standard metric; however, its binary nature often provides insufficient gradient signal, particularly for smaller LLMs that struggle to generate perfect queries in early training stages.

To mitigate this sparsity, we introduce an execution-based dense reward function based on the QATCH framework (Papicchio et al., 2023), utilizing *Cell Precision*, *Cell Recall*, and *Tuple Cardinality*. Our reward framework is structured into: (i) **atomic components**, which measure fundamental query properties, and (ii) **composite signals**, which synthesize these properties into a scalar optimization objective. While we provide the intuition below, the formal mathematical definitions are detailed in § D.

**Atomic Reward Components.** We categorize rewards into *format* and *task-specific* components. Following DeepSeek-AI et al. (2025), the format reward $R_{\text{FR}} \in \{0,1\}$ validates if the output adheres to the `<reasoning>` and `<answer>` tag structure. Instead, to evaluate task performance, we contrast sparse and dense feedback based on the execution results of the target query $\mathcal{T}$ and predicted query $\mathcal{T}_{\text{pred}}$:

*Execution Accuracy ($R_{EX} \in \{0,1\}$):* A sparse binary signal where $R_{\text{EX}} = 1$ iff $\mathcal{T} = \mathcal{T}_{\text{pred}}$, accounting for row multiplicity but invariant to column order.

*QATCH Reward ($R_{QA} \in [0,1]$):* A dense signal providing granular feedback via the mean of: (i) *Cell Precision* (CP), (ii) *Cell Recall* (CR), and (iii) *Tuple Cardinality* (TC), defined as $\min(|\mathcal{T}|, |\mathcal{T}_{\text{pred}}|)/\max(|\mathcal{T}|, |\mathcal{T}_{\text{pred}}|)$.

While $R_{\text{EX}}$ requires absolute correctness, $R_{\text{QA}}$ facilitates optimization through partial matches. For example ( Fig. 2), a query with an extra column but correct rows would yield $R_{\text{EX}} = 0$ but a non-zero $R_{\text{QA}}$ (e.g., $CP = 0.5, CR = 1.0$), guiding the model toward the correct solution even when the binary signal is absent. Note that $R_{\text{EX}} = 1 \implies R_{\text{QA}} = 1$.

**Composite Reward Signals.** We synthesize atomic components into a scalar signal using two methodologies: a weighted sum and a conditional gated reward formulation.

*Weighted Sum Formulation.* The baseline reward $R(\phi)$ is a weighted combination of task-specific and format metrics:

$$R(\phi) = 0.95 \cdot R_\phi + 0.05 \cdot R_{\text{FR}} \tag{2}$$

where $\phi \in \{\text{EX}, \text{QA}\}$, resulting in configurations $R_{\text{EXFM}}$ and $R_{\text{QAFM}}$. This weighting prioritizes SQL correctness while keeping the total reward within $[0, 1]$ to ensure training stability and prevent gradient spikes.

*Gated Execution Reward ($R_{GATE}$)* To decouple structural compliance from logical accuracy, $R_{\text{GATE}}$ revises $R_{\text{QAFM}}$ by penalizing non-executable queries and removing constant formatting noise for high-quality outputs. The reward $R_{\text{GATE}} \in \{0, 0.1\} \cup (0.1, 1]$ is defined as:

$$R_{\text{GATE}} = \begin{cases} 0 & \text{if query is non-executable} \\ 0.1 & \text{if executable, } R_{\text{QA}} \le 0.1, \text{ and } R_{\text{FR}} = 1 \\ R_{\text{QA}} & \text{if executable and } R_{\text{QA}} > 0.1 \end{cases} \tag{3}$$

This setup imposes a hard constraint on SQL syntax (reward 0) while providing a minimum gradient (floor 0.1) for validly formatted, logically weak queries. For higher-performing queries, the signal transitions purely to QATCH quality. This approach simplifies optimization by eliminating the constant formatting reward from the reward signal.

## 4 Experiments

In this section, we provide a systematic empirical evaluation of how reasoning capabilities affect Text-to-SQL performance across the model-scaling curve for the `Qwen3` family. Our investigation involves an extensive analysis of approximately 20,000 GPU hours.

The rest of the section is organized as follows. Section 4.1 details the experimental setup. Section 4.2 investigates **RQ1: Scale-Density Interdependence** and compares `Think2SQL` models against state-of-the-art baselines. Section 4.3 addresses **RQ2: Exploration Dynamic** and Section 4.4 draws conclusions on **RQ3: The Pareto Frontier.**

### 4.1 Experiments Setup

**Training setup.** We use the `Qwen3` family, trained with Open-R1 (HuggingFace, 2025; von Werra et al., 2020), with a fixed seed of 42. The `SFT` models are trained for 5 epochs on 4×H100 GPUs for 1h. We use the distilled reasoning dataset ($\mathcal{D}_{SFT}$), which comprises 5,682 samples. We hold out 20% (1,136 samples) for validation to trigger early stopping (patience = 3). The RLVR models are trained with DAPO (Yu et al., 2025a) without the dynamic sampling component for 1 epoch (batch size 256, micro-batch 64, learning rate $1 \times 10^{-6}$, 16 generations/batch) on 8×H100 80GB GPUs (4 update, 4 generation) over 15h (4B), 20h (8B), and 25h (14B) using BIRD training dataset (565 gradient steps) and vLLM (Kwon et al., 2023) with FlashAttention (Dao et al., 2022), temperature 0.6, top-$p$ 0.95 max generation token 4096 and repetition-penalty 1.1; more details in § C. Cold-start models follow DeepSeek-R1 (DeepSeek-AI et al., 2025) and have training times similar to RLVR. All models use a common prompt (§ C) and an enriched database schema (including descriptions and column examples, as in Li et al. (2025)). Best-performing models on BIRD-DEV are denoted `Think2SQL-4B`, `Think2SQL-8B`, and `Think2SQL-14B`.

**Model Baselines.** We evaluate a comprehensive suite of open- and closed-source models, encompassing both standard and reasoning-optimized architectures, in a zero-shot setting. To ensure a fair comparison, all general-purpose models are evaluated using a unified prompt structure adapted from Li et al. (2025), whereas specialized models such as `DeepRetrieval` (Jiang et al., 2025) are evaluated using their official task-specific templates.

Inference configurations follow official documentation to maximize model-specific performance. For the `Qwen3` family, we distinguish between the *Base* variant (non-reasoning model with *enable_thinking = False*, $\tau = 0.7$, top-$p = 0.8$, top-$k = 20$) and the *Thinking* variant (reasoning model with *enable_thinking = True*, $\tau = 0.6$, top-$p = 0.95$, top-$k = 20$). The *Base* variant serves as the default when not specified. For models lacking specific sampling guidelines, such as the `Llama 3.1` series (Grattafiori et al., 2024), we employ greedy decoding to ensure reproducibility of results. All reported std values, denoted as mean$_{\pm std}$, come from repeated stochastic decoding of a fixed checkpoint, not independent training reruns.

Open-source inference is performed using the `vLLM` engine, maintaining environmental parity with our RLVR training setup. Proprietary models, including `Gemini3-Flash`, and `GPT-5.2`, are accessed through their API. Due to the high computational cost ($\sim 300$ dollars for the entire evaluation set and for all API calls) and rate limits associated with reasoning-intensive endpoints, these models are evaluated in a single-pass setting.

Table 1: **Statistics of the evaluation datasets.** BIRD is our primary training/dev target, while the others serve as zero-shot generalization benchmarks.

| Category | Dataset | Instances | Primary Challenge |
|---|---|---:|---|
| Primary | BIRD (DEV) (Li et al., 2024b) | 1,530 | Real-world complexity & Scale |
| Robustness | SPIDER (TEST) (Yu et al., 2018) | 2,147 | Cross-domain generalization |
| | SPIDER-SYN (Gan et al., 2021a) | 1,034 | Lexical/Synonym shifts |
| | SPIDER-DK (Gan et al., 2021b) | 535 | Implicit domain knowledge |
| | SPIDER-REALISTIC (Deng et al., 2021) | 508 | Explicit column mention removal |
| Specialized | EHRSQL (DEV) (Lee et al., 2022) | 1,008 | Medical domain reasoning |

**Datasets and metrics.** We evaluate our models on a broad spectrum of benchmarks to assess performance on complex queries and robustness to distribution shifts, with summary statistics provided in Table 1. To evaluate specific reasoning challenges, we first utilize **BIRD** to test schema grounding at scale and **Spider** to measure compositional generalization across 200+ unseen databases. To further stress-test model robustness, we incorporate several Spider variants: **Spider-Syn** evaluates lexical robustness via synonym substitution, **Spider-DK** probes for implicit domain knowledge, and **Spider-Realistic** assesses ambiguity resolution through the removal of explicit mapping cues. Finally, we include **EHRSQL** to verify specialized domain adaptation in the high-stakes medical domain, which involves intricate temporal queries and technical terminology.

For performance quantification, we use the **refined execution accuracy**; details are provided in § A. Unlike standard execution accuracy (Li et al., 2024b), the refined version is: (i) *order-invariant*, permitting permutations in column selection that do not alter information content; and (ii) *set-sensitive*, strictly validating row counts and `DISTINCT` constraints. For example, given the target `SELECT` Name, Surname `FROM` Player;, our refined metric correctly accepts `SELECT` Surname, Name `FROM` Player; (a harmless column reordering that standard EX penalizes) and correctly rejects `SELECT DISTINCT` Name, Surname `FROM` Player; (a row-count alteration that standard EX overlooks). We adopt the refined execution accuracy throughout unless otherwise noted.

## 4.2 Main Results

This section details our empirical evaluation of the RLVR framework for the `Qwen3` family (4B, 8B, and 14B parameters). We begin by addressing **RQ1**, examining how reward density and scaling strategies synergize across different model capacities. We then evaluate the generalizability of our approach across multiple SQL benchmarks and conclude with a head-to-head comparison against current state-of-the-art (SOTA) proprietary and open-source models.

**Impact of Execution-guided dense rewards.** To address **RQ1**, we analyze the efficacy of our proposed execution-guided dense rewards against alternative formulations. Table 2 summarizes the performance of `Qwen3-4B` (*Base* and *Thinking* modes) trained via RLVR under five reward configurations: our proposed dense rewards ($R_{\text{GATE}}$ and $R_{\text{QAFM}}$), the standard sparse reward ($R_{\text{EXFM}}$), and two state-of-the-art baselines: $R_{\text{SQL-R1}}$ (Ma et al., 2025) and $R_{\text{Arctic-SQL}}$ (Yao et al., 2025). For the full results with standard deviation across three runs, see § E. Instead for the results under the standard *EX* metric see § A.

We observe four key trends: (i) RLVR consistently yields performance gains over both *Base* and *Thinking* variants. (ii) Reward density correlates positively with performance; our dense $R_{\text{GATE}}$ formulation achieves a 68.7% weighted average, outperforming the sparse $R_{\text{EXFM}}$ by +7% overall and +10% on BIRD-DEV, suggesting that smaller models benefit significantly from denser signals. (iii) Both $R_{\text{GATE}}$ and $R_{\text{QAFM}}$ outperform $R_{\text{Arctic-SQL}}$ and $R_{\text{SQL-R1}}$ by +1.5% and +2.7% in weighted averages and by +6% and +5% for BIRD-DEV, confirming the superiority of our execution-guided approach. (iv) $R_{\text{GATE}}$ specifically improves upon $R_{\text{QAFM}}$ on BIRD-DEV (59.6 vs 55.8), indicating that providing a "floor" bonus for executable queries while removing formatting noise enhances training convergence for `Qwen3-4B`.

Table 2: **Sensitivity Analysis of RLVR to Reward Function Design.** Evaluation of different reward formulations for RLVR training. `Qwen-4B` models are trained on BIRD-TRAIN and evaluated on BIRD-DEV, SPIDER variants and EHRSQL. We report the weighted average as a general performance metric with weights proportional to dataset sample sizes. *Base* and *Thinking* baselines represent the `Qwen3-4B` model prior to RLVR training in its standard and reasoning configurations, respectively. $R_{\text{Arctic-SQL}}$ (2025) and SQL-R1 (2025) are included as state-of-the-art reward comparisons.

| Model | BIRD dev #1,530 | Spider (test) #2,147 | Spider-DK #1,034 | Spider-Syn #535 | Spider-Realistic #508 | EHRSQL (dev) #1,008 | Weighted AVG |
|---|---|---|---|---|---|---|---|
| Base | $49.0_{\pm 0.5}$ | $78.6_{\pm 0.6}$ | $68.3_{\pm 1.5}$ | $68.0_{\pm 1.7}$ | $76.5_{\pm 0.9}$ | $29.0_{\pm 1.9}$ | 61.9 |
| Thinking | $52.0_{\pm 0.5}$ | $80.2_{\pm 0.2}$ | $70.7_{\pm 0.5}$ | $71.4_{\pm 1.0}$ | $78.8_{\pm 1.0}$ | $30.3_{\pm 0.8}$ | 64.1 |
| $R_{\text{EXFM}}$ | $54.5_{\pm 0.7}$ | $81.1_{\pm 0.1}$ | $70.4_{\pm 1.3}$ | $70.5_{\pm 0.1}$ | $77.9_{\pm 0.6}$ | $28.1_{\pm 1.4}$ | 64.5 |
| $R_{\text{QAFM}}$ | $55.8_{\pm 0.1}$ | $\mathbf{85.5_{\pm 0.4}}$ | $76.0_{\pm 1.1}$ | $\mathbf{75.5_{\pm 0.7}}$ | $\mathbf{82.5_{\pm 0.7}}$ | $32.8_{\pm 0.4}$ | 68.5 |
| $R_{\text{GATE}}$ | $\mathbf{59.6_{\pm 0.3}}$ | $83.7_{\pm 0.2}$ | $\mathbf{76.0_{\pm 0.3}}$ | $75.0_{\pm 0.5}$ | $81.2_{\pm 0.1}$ | $\mathbf{33.5_{\pm 0.3}}$ | $\mathbf{68.7}$ |
| $R_{\text{Arctic-SQL}}$ | $56.3_{\pm 1.2}$ | $83.1_{\pm 0.5}$ | $74.9_{\pm 0.8}$ | $75.5_{\pm 1.2}$ | $82.4_{\pm 0.4}$ | $33.2_{\pm 0.3}$ | 67.7 |
| $R_{\text{SQL-R1}}$ | $57.0_{\pm 1.1}$ | $83.4_{\pm 0.4}$ | $72.2_{\pm 0.8}$ | $72.3_{\pm 0.3}$ | $80.2_{\pm 0.7}$ | $31.7_{\pm 1.4}$ | 66.9 |

**Impact of Scaling and Model Capacity.** Fig. 3 illustrates the synergy between advantage scaling, reward density, and model capacity (4B, 8B, 14B) on BIRD-DEV (left) and the weighted average on SPIDER Variants and EHRSQL (right). Our analysis yields three primary insights:

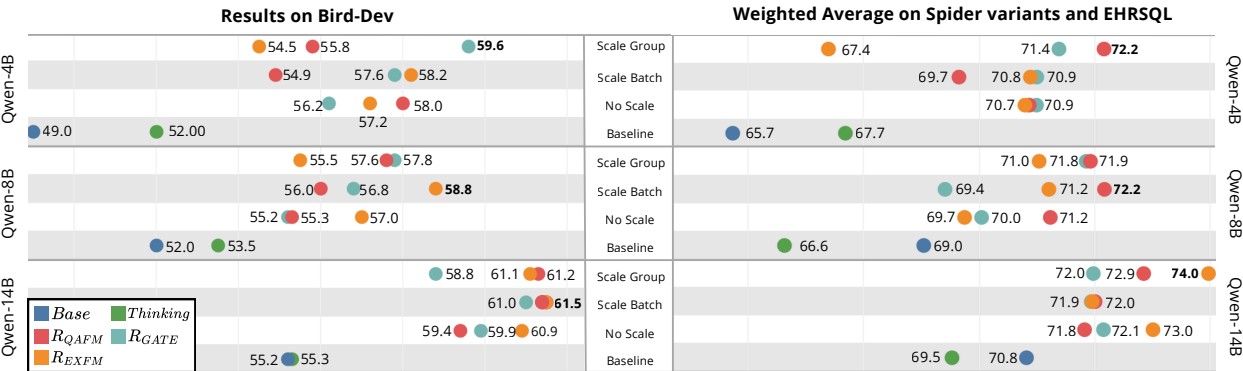

Figure 3: **Scaling Laws and Reward Normalization Dynamics in RLVR.** Comparative performance (EX) of RLVR-trained `Qwen3` models (4B, 8B, and 14B) under different advantage normalization strategies (*No Scale*, *Group Scaling*, and *Batch Scaling*) and reward formulations ($R_{QAFM}$, $R_{GATE}$, and $R_{EXFM}$). Models are trained on BIRD-TRAIN and evaluated on BIRD-DEV (Left) and SPIDER variants and EHRSQL reported as weighted average (Right), with weights proportional to dataset sample sizes. *Base* and *Thinking* represent `Qwen-4B` in its standard and reasoning configurations.

First, *model scale is the primary determinant of robustness.* As capacity increases to 14B, the performance gap between various scaling strategies and reward formulations narrows significantly compared to the 4B regime. This suggests that larger models possess an inherent architectural resilience to reward sparsity and group scaling.

Second, *Group Scaling facilitates peak performance but requires signal density at small scales.* Group Scaling mostly achieves the highest accuracy by leveraging lower-variance relative advantages. However, it is highly sensitive to reward sparsity; we observe a 10% performance delta between $R_{EXFM}$ (sparse) and $R_{GATE}$ (dense) at the 4B scale (54.5 vs 59.6). Dense rewards are particularly helpful for smaller models; one plausible explanation is that they reduce the incidence of zero-variance groups during training, see § F for details . In contrast, *Scale Batch* and *No Scale* configurations exhibit greater robustness to sparse signals, making $R_{EXFM}$ competitive for larger models (58.8 for 8B and 61.3 for 14B).

Third, we identify a *diminishing return of the dense "floor" bonus.* The "floor" bonus provided by $R_{GATE}$, which stabilizes the 4B model, becomes a liability in the 14B regime. In high-capacity models, standard $R_{QAFM}$ and even sparse $R_{EXFM}$ begin to outperform $R_{GATE}$. This indicates that the dense "floor" may encourage reward hacking or the acquisition of suboptimal shortcuts in models with sufficient capacity to exploit such signals, see § F for more details.

**Summary for RQ1:** Our findings suggest that maintaining a consistent, high-fidelity reward signal is essential. For smaller models (4B), this is best achieved via dense rewards paired with Group Scaling. Conversely, for larger models (8B+), the risk of overfitting to dense proxies grows; here, sparse rewards combined with aggressive Batch Scaling mostly yield superior generalization.

**Main Results and Comparative Analysis.** To translate our findings to competitive performance, we evaluate the full `Think2SQL` suite (4B, 8B, and 14B) against a diverse set of SOTA baselines (Table 3). `Think2SQL` configurations are selected by peak BIRD-DEV performance, using out-of-distribution weighted averages as a tie-breaker for results within one standard deviation (see § E). The `Think2SQL` setups use $R_{GATE}$ with Group Scaling (4B) and $R_{EXFM}$ with Batch Scaling (8B) and Group Scaling (14B).

Instead, the competitors are categorized into: (i) RL-tuned reasoning models (SQL-R1); (ii) SFT-based specialized models `OmniSQL`; and (iii) frontier general-purpose LLMs. We exclude `Arctic-SQL` from direct comparison as its training set includes SPIDER (Yao et al., 2025), which would compromise the validity of the zero-shot generalization analysis on Spider variants. To maximize fairness, specialized models are evaluated using the prompts and settings from their original papers, whereas general-purpose models use a standardized

Table 3: **Main Results: Execution Accuracy (EX) across Text-to-SQL Benchmarks.** Performance of `Think2SQL` vs. baselines on BIRD, SPIDER variants, and EHRSQL. Weighted average (Weigh. AVG) is weighted on dataset sample size. Subscripts (▲) denote standard deviation ($n = 3$) and green markers denote relative RLVR gains (%) over the *Base* backbone. Bold values indicate best-in-class performance. `o4-mini` and `Cod` denote specific `o4-mini-2025-04-16` and `Coder` versions. The ✓ denotes models specifically optimized for Text-to-SQL or reasoning via RLVR/specialized training, whereas the ✗ denotes general-purpose instruct models.

| LLM | BIRD (dev) #1,530 | Spider test #2,147 | Spider-DK #1,034 | Spider-Syn #535 | Spider-Realistic #508 | EHRSQL #1,008 | Weigh. AVG |
|---|---|---|---|---|---|---|---|
| **Open-source LLMs ($< 5$B)** | | | | | | | |
| ✗ Qwen3-0.6B | $15.4_{+0.4}$ | $45.5_{+0.8}$ | $40.9_{+1.5}$ | $40.4_{+0.2}$ | $41.6_{+1.1}$ | $8.6_{+0.4}$ | 31.8 |
| ✗ Qwen3-1.7B | $33.4_{+0.9}$ | $69.2_{+0.1}$ | $59.0_{+1.2}$ | $60.1_{+0.2}$ | $66.0_{+1.1}$ | $12.9_{+0.3}$ | 50.2 |
| ✗ Qwen2.5-Cod-3B | 33.4 | 67.4 | 57.2 | 53.7 | 56.5 | 16.4 | 48.6 |
| ✓ DeepRetrieval-3B | $42.2_{+0.4}$ | $77.0_{+0.3}$ | $67.5_{+0.5}$ | $67.2_{+0.9}$ | $72.5_{+0.9}$ | $21.9_{+0.5}$ | 58.3 |
| ✓ SQL-R1-3B | $42.9_{+0.7}$ | $76.3_{+0.4}$ | $66.0_{+0.7}$ | $63.1_{+0.9}$ | $66.2_{+0.6}$ | $31.8_{+0.2}$ | 58.7 |
| ✗ Qwen3-4B | $49.0_{+0.5}$ | $78.6_{+0.6}$ | $68.3_{+1.5}$ | $68.0_{+1.7}$ | $76.5_{+0.9}$ | $29.0_{+1.9}$ | 61.9 |
| ✓ **Think2SQL-4B** | $\mathbf{59.6_{+0.3}}$ ▲22 | $\mathbf{83.7_{+0.2}}$ ▲6 | $\mathbf{76.0_{+0.3}}$ ▲9 | $\mathbf{75.0_{+0.5}}$ ▲9 | $\mathbf{81.2_{+0.1}}$ ▲6 | $\mathbf{33.5_{+0.3}}$ ▲16 | **68.7** |
| **Open-source LLMs (5-10B)** | | | | | | | |
| ✓ DeepRetrieval-7B | $49.9_{+0.1}$ | $80.2_{+0.6}$ | $72.3_{+0.5}$ | $68.3_{+08}$ | $76.2_{+0.3}$ | $29.9_{+0.6}$ | 63.4 |
| ✗ Qwen2.5-Cod-7B | 43.6 | 80.3 | 68.1 | 69.3 | 76.5 | 20.2 | 60.0 |
| ✗ OmniSQL-7B | 57.0 | 85.4 | 75.8 | 74.4 | 81.0 | **36.9** | 69.1 |
| ✓ SQL-R1-7B | $58.8_{+0.5}$ | $\mathbf{86.2_{+0.1}}$ | $74.4_{+0.6}$ | $74.9_{+0.3}$ | $\mathbf{81.2_{+0.5}}$ | $35.4_{+0.4}$ | **69.4** |
| ✗ Llama3.1-8B | 39.7 | 72.8 | 63.8 | 64.1 | 70.9 | 23.1 | 55.7 |
| ✗ Qwen3-8B | $52.0_{+0.6}$ | $81.9_{+0.2}$ | $71.7_{+1.5}$ | $71.4_{+0.8}$ | $79.4_{+0.2}$ | $31.6_{+0.7}$ | 65.1 |
| ✓ **Think2SQL-8B** | $58.8_{+0.4}$ ▲14 | $83.7_{+0.5}$ ▲3 | $\mathbf{75.9_{+0.7}}$ ▲6 | $74.9_{+0.6}$ ▲5 | $80.9_{+0.6}$ ▲2 | $32.9_{+0.4}$ ▲2 | 68.4 |
| **Open-source LLMs (10-50B)** | | | | | | | |
| ✓ SQL-R1-14B | $43.2_{+1.0}$ | $73.5_{+0.7}$ | $64.5_{+0.8}$ | $65.9_{+1.1}$ | $70.0_{+2.9}$ | $30.6_{+0.6}$ | 58.0 |
| ✗ Qwen3-14B | $55.2_{+0.4}$ | $83.9_{+0.4}$ | $74.0_{+1.4}$ | $73.2_{+0.5}$ | $81.3_{+0.6}$ | $32.0_{+0.8}$ | 67.1 |
| ✗ OmniSQL-14B | 58.6 | 86.2 | 75.0 | 75.0 | 82.4 | 35.5 | 69.5 |
| ✗ Qwen3-Cod-30B | $61.0_{+0.3}$ | $\mathbf{86.9_{+0.3}}$ | $76.8_{+0.1}$ | $78.3_{+0.6}$ | $\mathbf{84.3_{+0.6}}$ | $37.5_{+1.1}$ | 71.1 |
| ✗ Qwen3-32B | $53.9_{+0.4}$ | $84.9_{+0.1}$ | $74.4_{+1.4}$ | $74.6_{+0.3}$ | $83.2_{+0.5}$ | $32.8_{+0.8}$ | 67.6 |
| ✗ Qwen2.5-Cod-32B | 57.3 | 86.1 | 76.9 | 77.6 | 84.0 | 35.4 | 69.8 |
| ✗ OmniSQL-32B | 58.6 | 86.4 | 76.4 | 76.8 | 82.2 | **39.1** | 70.5 |
| ✓ **Think2SQL-14B** | $\mathbf{61.3_{+0.8}}$ ▲11 | $85.6_{+0.4}$ ▲2 | $\mathbf{77.8_{+0.9}}$ ▲5 | $\mathbf{78.6_{+0.1}}$ ▲7 | $81.8_{+0.6}$ ▲1 | $38.9_{+0.3}$ ▲20 | **71.3** |
| **Closed-Source LLMs** | | | | | | | |
| ✗ GPT-5.2 | 59.0 | 81.5 | 68.0 | 71.9 | 76.0 | 43.1 | 67.4 |
| ✗ o3-2025-04-16 | 59.8 | 81.4 | 67.1 | 72.1 | 77.8 | 40.8 | 67.3 |
| ✗ o4-mini | 58.4 | 83.3 | 73.8 | 74.6 | 84.3 | 43.8 | 69.7 |
| ✓ Gemini3-flash | **65.2** | 89.3 | **80.7** | 81.5 | **87.6** | 40.2 | 74.5 |
| ✓ Gemini3-Pro | **65.2** | **89.5** | 78.1 | **84.5** | 87.4 | **41.9** | **74.6** |

prompt format shown to be effective for Text-to-SQL (Li et al., 2025). In the following results, pp denotes percentage points (the absolute difference between two scores), and % denotes relative improvement.

*Think2SQL consistently outperforms Text-to-SQL-specialize baselines across all scales.* At the 4B scale, `Think2SQL-4B` achieves 59.6 EX on BIRD-DEV, a +22% relative improvement (+10.6 pp) over the `Qwen3-4B` base. Notably, it outperforms the leading RL-based competitor, `SQL-R1-3B` (42.9 EX), by 16.7 pp, suggesting that our RLVR framework effectively bridges the gap between latent reasoning and structured code generation. At the 8B scale, `Think2SQL-8B` reaches 58.8 EX, a +14% relative gain over its base. While `SQL-R1-7B` reports a higher weighted average (69.4 vs 68.4 for `Think2SQL-8B`), it requires a data-intensive two-step pipeline (200k SFT and 5k RLVR samples); In contrast, our approach achieves near-parity without this extensive SFT overhead, resulting in significantly more data-efficient training while trading off a small amount of absolute peak performance. This trade-off is deliberate: our goal is controlled empirical characterization under comparable budgets, not exhaustive scaling to maximize leaderboard performance at each model size. Finally, at the 14B scale, `Think2SQL-14B` attains 61.3 EX (+11% relative gain over the `Qwen3-14B` base), surpassing

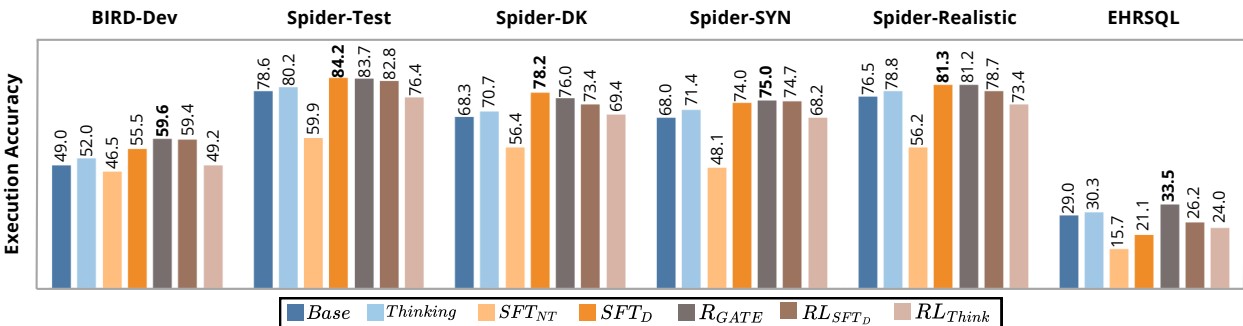

Figure 4: **Cold-Start impact on RLVR Performance.** Performance comparison of different training strategies for `Qwen3-4B` trained on BIRD-TRAIN. All RLVR experiments use the $R_{\text{GATE}}$ configuration with group scaling. *Base* and *Thinking* denote `Qwen-4B` in base and Thinking configuration; $SFT_{NT}$ denotes the SFT without reasoning traces. $SFT_D$ denotes the SFT on our constructed datasets $\mathcal{D}_{SFT}$. $RL_{SFT_D}$ denotes RLVR starting from the SFT-trained model $SFT_D$. $RL_{\text{Think}}$ denotes RLVR starting from the *Thinking* configuration.

`OmniSQL-32B` (58.6 EX) by 2.7 pp despite having less than half the parameter count. This confirms that the efficacy of our RLVR approach scales robustly with model capacity while maintaining a clear efficiency advantage.

*Parity with Proprietary Frontier Models.* `Think2SQL` in all scales demonstrates competitive performance against leading proprietary models. Notably, `Think2SQL-4B` (59.6 EX on BIRD-DEV) outperforms `GPT-5.2` (59.0 EX) and rivals the reasoning-heavy `o3-2025` (59.8 EX), despite being an order of magnitude smaller in parameter count. Similarly, `Think2SQL-14B` (61.3 EX on BIRD-DEV and 71.2 on average) surpasses GPT and `o` variants and closely approaches `Gemini3` family ($\sim 74.5$ on average).

## 4.3 The Cold Start Problem

A critical bottleneck for Reinforcement Learning from Verifiable Rewards (RLVR) is the quality of the initial policy, the *cold start problem* (**RQ2**). To investigate how different inductive biases in the initialization phase affect final RLVR convergence and generalization, we compare three training strategies on BIRD-TRAIN using `Qwen3-4B`: (i) Zero-Shot Learning (ZSL): *Base* model without task-specific fine-tuning or its *Thinking* configuration; (ii) Supervised Fine-Tuning (SFT): the $SFT_D$ model, fine-tuned on synthetic reasoning traces distilled from `Gemini3-Flash`; and the $SFT_{NT}$ model, fine-tuned on SQL outputs without reasoning traces; (iii) the Reinforcement Learning with Verifiable Reward (RLVR) starting from each of the above initializations: $RL_{Base}$, $RL_{SFT_D}$, and $RL_{\text{Thinking}}$. The RLVR is configured to achieve the highest performance gains: `Qwen3-4B` with $R_{\text{GATE}}$ and group scaling.

Fig. 4 illustrates the performance across BIRD-TEST, SPIDER variants, and EHRSQL. For the full results with standard deviation across three runs, see § E. We can define two key insights from the results:

*Reasoning vs. Non-Reasoning SFT.* Our results indicate that fine-tuning without explicit reasoning traces ($SFT_{NT}$) is detrimental, often underperforming the base model. This suggests that "shallow" SFT on SQL outputs alone may diminish the model's general-purpose reasoning capabilities. Conversely, the $SFT_D$ provide substantial gains over the *Base* and *Thinking* configurations, suggesting that distillation from stronger teacher models, such as `Gemini3-Flash`, effectively injects domain-specific reasoning patterns. However, the performance gap between $SFT_D$ and the base model narrows or reverses on EHRSQL, supporting the hypothesis that SFT-based reasoning is prone to *distributional mimicry* rather than robust logic, leading to fragile generalization in out-of-domain schemas (Bender et al., 2021).

*The RLVR Cold Start Bias.* Comparing RLVR outcomes across initializations, we observe a significant *initialization bias*. Counterintuitively, RLVR, when initialized from the **Base model**, often achieves the

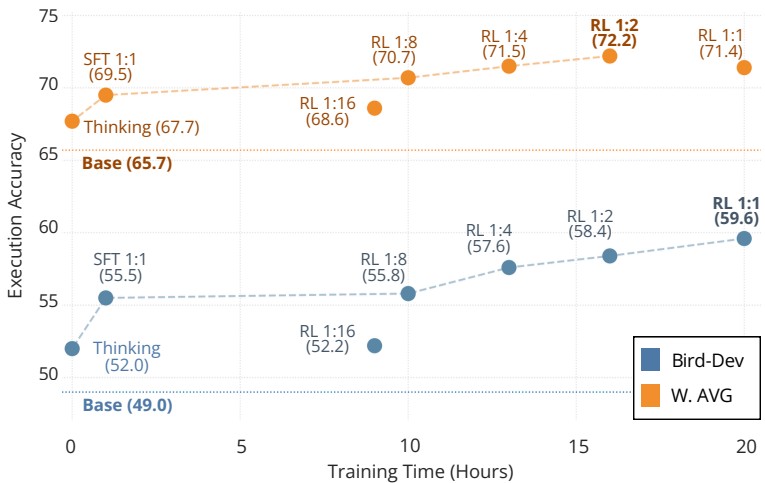

Figure 5: **Pareto Frontier of Execution Accuracy vs. Training Budget.** Scaling trajectories of model performance as a function of total wall-clock training time. Models are trained on BIRD-TRAIN and evaluated on BIRD-DEV (blue) and SPIDER variants and EHRSQL reported as Weighted Average (W. AVG) (orange) with weights proportional to dataset size. For RLVR regimes, the number of gradient updates is fixed ($\sim$565), while the training budget is modulated by varying the dataset subsampling ratio ($RL_{1:n}$). $SFT_{1:1}$ refers to the SFT-trained Qwen-4B on our collected datasets $\mathcal{D}_{SFT}$. *Base* and *Thinking* refer to Qwen-4B in *Base* and *Thinking* mode, respectively.

highest peak performance and the most robust generalization. In contrast, starting from *Thinking* ($RL_{Think}$) or $SFT_D$ ($RL_{SFT_D}$) frequently results in sub-optimal convergence. We hypothesize that pre-configuring the model with specific reasoning traces constrains the exploration space. While these initializations provide a higher starting reward, they may bias optimization toward narrower regions of the policy space (see § F for details).

**Summary for RQ2:** Our findings suggest that (i) for complex reasoning tasks, RLVR is more effective than supervised alignment with or without reasoning traces; and (ii) the cold start configuration dictates the ceiling of RLVR performance. While SFT backbones provide an immediate performance boost, they impose an *exploration tax* that leads to sub-optimal convergence.

## 4.4 Data Efficiency and Temporal Overhead

To investigate the Pareto frontier between wall-clock time and model performance (**RQ3**), we conduct an ablation on data efficiency. A critical bottleneck in RLVR for Text-to-SQL is the **execution-induced latency**. Unlike standard SFT, calculating the reward $\mathcal{R}$ requires executing model-generated queries $\hat{y}$ against a DBMS. This introduces a computational overhead of $\mathcal{O}(M \cdot S)$, where $M$ is the number of samples and $S$ denotes schema-linking complexity, resulting in a significant wall-clock time divergence between RLVR and SFT.

We focus our analysis on the RLVR configuration that leads to the highest performance gains: Qwen3-4B with $R_{\text{GATE}}$ and group scaling. We evaluate the Qwen3-4B trained under various subsampled regimes, denoted as $RL_{1:n}$, where the dataset $\mathcal{D}_n \subset \mathcal{D}$ is reduced by a factor of $n$. To isolate the impact of sample diversity from the total optimization signal, we fix the budget of gradient updates ($K \approx 565$) by adjusting the number of training epochs $E_n = \left\lfloor \frac{n \cdot K \cdot B}{|\mathcal{D}_n| \cdot G} \right\rfloor$ where $B$ is the effective batch size (256) and $G$ is the number of generations per sample (16). In our setup, this simplifies to $E_n = n$, thereby maintaining a constant number of gradient updates across regimes. The formula is derived from the total number of optimization steps $K = \frac{|\mathcal{D}_n| \cdot G \cdot E_n}{n \cdot B}$. In Fig. 5, we compare the subsampled regimes with the SFT-trained model on our collected datasets $\mathcal{D}_{SFT}$ and the *Base* and *Thinking* configurations of Qwen-4B. For the full results with standard deviation across three runs, see § E. Reported wall-clock times cover the training phase only. For RLVR, this includes

reward computation (executed on CPU) and the policy update loop, using 4×H100 GPUs for vLLM rollout generation and 4×H100 GPUs for policy updates. For SFT, it includes the standard forward and backward passes on 4×H100 GPUs.

*Sublinear Scaling of Temporal Overhead.* While the full dataset ($RL_{1:1}$) requires 20 hours, the $RL_{1:16}$ regime completes in under 9 hours. This 55% reduction in wall-clock time, despite a 94% reduction in unique samples, indicates that fixed costs in environment initialization and distributed synchronization dominate the training pipeline as $n$ increases.

*Sample Efficiency as Distributional Alignment.* RLVR demonstrates significant resilience to data scarcity. Notably, $RL_{1:2}$ ($\sim 4,500$ samples) achieves a weighted average EX of 72.2, surpassing the full-data configuration (71.4). This suggests that RLVR primarily functions as a *distributional aligner* for pre-existing knowledge (Yue et al., 2025). We hypothesize that moderate subsampling acts as a stochastic regularizer, preventing the policy from over-fitting to schema-specific noise and improving generalization to unseen database structures.

*RLVR vs. SFT Pareto Dominance.* Even at 12.5% data capacity ($RL_{1:8}$), RLVR outperforms the SFT baseline. Although SFT remains 10× faster, its performance is upper-bounded by the quality of the teacher distribution. RLVR's ability to surpass SFT with minimal samples underscores its utility in domains where high-quality expert trajectories are unavailable but a deterministic reward signal is accessible. For extreme subsampling ($RL_{1:16}$), RLVR underperforms SFT, suggesting a lower bound on the data is required for effective reward-driven learning.

Predicted query caching across epochs presents an additional optimization lever. Since reward computation consumes the majority of the computational budget, memoizing execution results for duplicate SQL outputs can reduce per-update latency. We refer to future work for a systematic investigation of this caching strategy.

**Summary of RQ3:** Our analysis reveals that RLVR exhibits robust sample efficiency, with moderate data subsampling enhancing generalization through implicit regularization. While RLVR mostly outperforms SFT across the Pareto frontier of training time versus model quality, extreme subsampling can lead to underperformance relative to SFT. Nonetheless, RLVR remains highly effective in scenarios where expert demonstrations are scarce but verifiable rewards are obtainable.

## 5 Conclusion

In this paper, we presented a comprehensive empirical study examining the impact of reasoning capabilities into Text-to-SQL (RLVR). Our analysis systematically quantifies the interplay between reward granularity, advantage normalization, and model scale, establishing a reproducible framework for optimizing parameter-efficient architectures. We demonstrate that sparse reward signals yield suboptimal convergence for small models, whereas our execution-guided dense reward outperforms state-of-art rewards.

Moreover, our observations on distributional alignment reveal that conventional supervised fine-tuning may obscure latent reasoning gaps that RLVR effectively addresses. The `Think2SQL` model family validates that sophisticated reasoning is not intrinsically tied to model scale; our 4B variant achieves performance parity with substantially larger baselines, advancing the frontier of efficient specialized reasoning systems.

## 6 Limitations

Despite these gains, limitations remain. Our study focuses on the `Qwen3` architecture; generalizability to other families (e.g., Llama) requires further validation. Additionally, while our dense reward function outperforms current baselines, the search space for hybrid reward signals is not yet exhausted. While we adopt a unified prompt structure across models to ensure a controlled comparison, we note that individual baselines may benefit from model-specific prompt engineering. Finally, while our models excel on standard benchmarks, their robustness in noisy, real-world database environments remains an open challenge (Papicchio et al., 2025a). Future work will extend this blueprint to broader architectures and dynamic evaluation settings exploring a Dense-to-Sparse Reward Curriculum for training small LMs.

**Acknowledgments**

This project was provided with computer and storage resources by GENCI at IDRIS, thanks to grants 2025-AD010616649 and 2025-AD010616180 on the supercomputer Jean Zay's H100 and A100 partitions. This work has also been partially supported by the 3IA Côte d'Azur Investments in the IA-cluster project managed by the National Research Agency (ANR) with the reference number ANR-23-IACL-0001.

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

## A Execution accuracy vs Refined Execution accuracy

Standard coding of **Execution Accuracy (EX)** (Li et al., 2024b) frequently encounters semantic inconsistencies due to its reliance on naive set-based comparisons Algorithm 1. Specifically, existing implementations exhibit two primary shortcomings: (i) *Multiplicity Loss*, where the omission of bag semantics causes queries with and without `DISTINCT` to be treated as equivalent; and (ii) *Column Sensitivity*, where valid permutations of projected columns are incorrectly penalized.

As illustrated in Algorithm 1, let $r_y$ and $r_p$ denote the resulting tables of the executed ground-truth and predicted queries, respectively, represented as lists of tuples. Standard EX fails to normalize the internal tuple structure because the set-based comparison is applied only to the outer collection, leaving the metric vulnerable to the aforementioned issues. To resolve these limitations, we propose **Refined EX** (Algorithm 2), a metric that adopts bag semantics to preserve row multiplicity and enforces attribute-order invariance via canonical sorting of elements within each tuple prior to comparison. This refinement ensures that the evaluation captures the underlying informational content rather than superficial structural formatting.

To support reproducible and standardized benchmarking, we introduce NL2SQLEval[3], an extensible Python toolkit for modular SQL evaluation.

---

**Algorithm 1** BIRD_EX

1: **procedure** BIRD_EX$(y, x)$
2:     $r_y \leftarrow \text{exec}(y)$
3:     $r_p \leftarrow \text{exec}(x)$
4:     **if** $set(r_y) \neq set(r_p)$ **then**
5:         **return false**
6:     **return true**

---

**Algorithm 2** Refined_EX

1: **procedure** REFINED_EX$(y, x)$
2:     $r_y \leftarrow \text{exec}(y)$
3:     $r_p \leftarrow \text{exec}(x)$
4:     **if** $|r_y| \neq |r_p|$ **then**
5:         **return false**
6:         $r_y \leftarrow \text{SortTuples}(\text{SortIntraTuple}(r_y))$
7:         $r_p \leftarrow \text{SortTuples}(\text{SortIntraTuple}(r_p))$
8:     **for** $i \leftarrow 1$ to $|r_y|$ **do**
9:         **if** $r_y[i] \neq r_p[i]$ **then**
10:           **return false**
11:         **return true**

---

Note that while the absolute performance metrics shift across different reward configurations, the relative gains between models remain consistent. As shown between Tables 2 and 4, our findings are robust to variations in the evaluation protocol. Specifically, we observe that scores under classical Execution Accuracy (EX) are systematically higher for BIRD-DEV and lower for SPIDER variants. Qualitative analysis reveals that the gains primarily stem from the *Multiplicity Loss* problem, errors induced by the presence or absence of the `DISTINCT` keyword. This phenomenon is particularly pronounced in the BIRD-DEV benchmark. Instead, the lower results stem from the *Column Sensitivity* issue, where valid column permutations are unfairly penalized.

## B Fine-Tuning Language Models

### B.1 Supervised Fine-Tuning

Supervised Fine-Tuning (SFT) adapts a pretrained language model $\pi_{\boldsymbol{\theta}}$ to a distribution $\mathcal{P}$ of sequences that reflect desired linguistic or task-specific behavior. Let $\boldsymbol{z} = (z_1, z_2, \ldots, z_T)$ denote a token sequence drawn from $\boldsymbol{z} \sim \mathcal{P}$. The SFT objective maximizes the likelihood of sequences under $\pi_{\boldsymbol{\theta}}$, which corresponds to minimizing the expected negative log-likelihood:

$$\mathcal{L}_{\text{full}}(\boldsymbol{\theta}) = -\mathbb{E}_{\boldsymbol{z} \sim \mathcal{P}} \left[ \sum_{t=1}^{T} \log \pi_{\boldsymbol{\theta}}(z_t \mid \boldsymbol{z}_{<t}) \right]. \tag{4}$$

---

[3] https://github.com/spapicchio/NL2SQLEvaluator

Table 4: **Sensitivity Analysis of RLVR to Reward Function Design with classical EX (2024b).** Sensitivity analysis as in Table 2 but calculating EX with Algorithm 1. The relative performance gains are consistent with Table 2, confirming the robustness of our findings across different evaluation protocols.

| Model | BIRD dev #1,530 | Spider (test) #2,147 | Spider-DK #1,034 | Spider-Syn #535 | Spider-Realistic #508 | EHRSQL (dev) #1,008 | Weighted AVG |
|---|---|---|---|---|---|---|---|
| Base | $54.2^{(0.7)}$ | $75.5^{(0.5)}$ | $64.4^{(1.4)}$ | $64.4^{(1.4)}$ | $74.5^{(1.4)}$ | $30.4^{(1.9)}$ | 61.3 |
| Thinking | $57.6^{(0.6)}$ | $76.4^{(0.2)}$ | $65.7^{(0.7)}$ | $67.4^{(0.9)}$ | $75.8^{(1.5)}$ | $32.3^{(0.7)}$ | 63.2 |
| $R_{\text{EXFM}}$ | $58.9^{(0.8)}$ | $76.9^{(0.1)}$ | $65.8^{(0.4)}$ | $66.3^{(0.1)}$ | $75.4^{(0.4)}$ | $29.1^{(1.4)}$ | 63.1 |
| $R_{\text{QAFM}}$ | $60.3^{(0.1)}$ | $81.1^{(0.5)}$ | $70.4^{(1.0)}$ | $71.9^{(0.5)}$ | $79.9^{(0.5)}$ | $34.4^{(0.5)}$ | 67.0 |
| $R_{\text{GATE}}$ | $63.6^{(0.3)}$ | $80.0^{(0.3)}$ | $72.4^{(0.7)}$ | $72.6^{(0.7)}$ | $80.5^{(0.4)}$ | $33.7^{(0.3)}$ | **67.7** |
| $R_{\text{Arctic-SQL}}$ | $60.7^{(1.5)}$ | $79.4^{(0.5)}$ | $69.6^{(0.9)}$ | $71.4^{(1.2)}$ | $79.8^{(0.4)}$ | $34.8^{(0.3)}$ | 66.4 |
| $R_{\text{SQL-R1}}$ | $62.7^{(0.9)}$ | $79.7^{(0.3)}$ | $67.9^{(1.0)}$ | $68.6^{(0.1)}$ | $77.3^{(0.9)}$ | $33.7^{(1.3)}$ | 66.1 |

```
<reasoning>
Alright, let's break this down. My goal is to generate an SQL query [...]
First, I need to understand the data. [...]
My database knowledge, especially with text-to-SQL tasks, suggests [...]
Looking at the schema, I note the crucial join condition [...]
Finally, I add the filter condition `T2.Profit < 0`
 to only pick products which are showing profit deficiency.
</reasoning>

<answer>
To find the names of the products that have a profit deficiency (negative profit)
[...]
Steps:
1. Identify the goal: Get the `Product Name` from the `product` table.\n2.
2. Define "deficiency": According to the prompt, deficiency refers to [...].
3. Link the tables: Join `product` and `central_superstore` [...]
4. Filter the data: Apply the condition `Profit < 0` [...]
5. Select the output: Return the unique `Product Name` values.

```sql
SELECT DISTINCT [...]
```

</answer>
```

Figure 6: Example of a distilled reasoning trace from the BIRD training set with `Gemini3-Flash`. [...] indicates omitted content for brevity.

For tasks with an explicit decomposition into an input segment and a target segment—such as QA, summarization, or assistant-style dialogue—the data distribution consists of pairs $(\boldsymbol{x}, \boldsymbol{y}) \sim \mathcal{P}$, where $\boldsymbol{x} = (x_1, \ldots, x_n)$ is the conditioning prompt and $\boldsymbol{y} = (y_1, \ldots, y_m)$ is the supervised output. In such settings, the model conditions on $\boldsymbol{x}$ and predicts the continuation $\boldsymbol{y}$, with the loss computed only over the target tokens:

$$\mathcal{L}_{\text{cond}}(\boldsymbol{\theta}) = -\mathbb{E}_{(\boldsymbol{x}, \boldsymbol{y}) \sim \mathcal{P}} \left[ \sum_{t=1}^{m} \log \pi_{\boldsymbol{\theta}}(y_t \mid \boldsymbol{x} \| \boldsymbol{y}_{<t}) \right], \tag{5}$$

where $\|$ denotes the concatenation operator. This alternative objective is often preferred in practice, as it allows for more efficient training by focusing on the relevant output tokens and ignoring the input tokens

(Chiang et al., 2023; Yu et al., 2024; Wang et al., 2023). More recently, Shi et al. (2024) showed that models trained with the SFT objective in Eq. (4) can be superior to Eq. (5) when the target sequence is significantly shorter than the input sequence. In the case of distillation of reasoning models, the output sequence will be considerably longer than the input sequence, and the SFT objective in Eq. (5) is preferred. Finally, the expectations in Eq. (4) and Eq. (5) are approximated by empirical means over a finite dataset $\mathcal{D} = \{z_i\}_{i=1}^{N}$ or $\mathcal{D} = \{(x_i, y_i)\}_{i=1}^{N}$ consisting of $N$ training examples. The resulting objective is optimized via standard stochastic gradient descent or its variants (Robbins & Monro, 1951; Kingma & Ba, 2014).

This study employs SFT to adapt base LLMs to the Text-to-SQL task using a distilled dataset with `Gemini3-Flash`. Given that `Gemini3-Flash` provides a summarized abstraction of its internal reasoning rather than a raw chain-of-thought, we preserve both the distilled reasoning summary (enclosed in `</reasoning>` tags) and the generated SQL statement (enclosed in `</answer>` tags). An example of a distilled reasoning trace is shown in Fig. 6.

## B.2 Reinforcement Learning with Verifiable Rewards (RLVR)

Group-Relative Policy Optimization (GRPO) (Shao et al., 2024) has been recently introduced as a value-free alternative to Proximal Policy Optimization (PPO) (Schulman et al., 2017) for fine-tuning language models.

Let $x \sim \mathcal{X}$ denote a prompt drawn from a distribution over conditioning inputs, and let $\{y_i\}_{i=1}^{G} \sim \pi_{\theta_{\text{old}}}(\cdot \mid x)$ be $G$ response sequences generated by the frozen reference policy $\pi_{\theta_{\text{old}}}$. Each response $y_i = (y_{i,1}, \ldots, y_{i,T_i})$ is assigned a scalar reward $R_i \in \mathbb{R}$ computed via a reward function.

The group-relative advantage $A_i$ for the $i$-th response is defined by normalizing the reward distribution over the group:

$$A_i = \frac{R_i - \mathbb{E}[R_j]}{\sqrt{\mathbb{V}[R_j]}}, \qquad j \in \{1, \ldots, G\}, \tag{6}$$

where $\mathbb{E}[R_j]$ and $\mathbb{V}[R_j]$ are the mean and variance of the rewards for the group of responses, respectively. For each token position $t$ in response $y_i$, define the state as $s_{i,t} = x \| y_{i,<t}$, and the token-level probability ratio as

$$p_{i,t}(\theta) = \frac{\pi_\theta(y_{i,t} \mid s_{i,t})}{\pi_{\theta_{\text{old}}}(y_{i,t} \mid s_{i,t})}.$$

The GRPO training objective minimizes a clipped surrogate loss penalized by the KL divergence from the reference policy:

$$\mathcal{L}_{\text{GRPO}}(\theta) = \mathbb{E}\left[\frac{1}{G}\sum_{i=1}^{G}\frac{1}{T_i}\sum_{t=1}^{T_i}\min\left(p_{i,t}(\theta)A_i, \text{clip}(p_{i,t}(\theta), 1-\epsilon, 1+\epsilon)A_i\right) - \beta \text{KL}\left[\pi_\theta \| \pi_{\theta_{\text{ref}}}\right]\right] \tag{7}$$

where the expectation is taken over the prompt distribution $x \in \mathcal{X}$, and the responses $\{y_i\}_{i=1}^{G}$ generated by the frozen policy $\pi_{\theta_{\text{old}}}$. Additionally, $\epsilon$ is set to be the clipping parameter and $\beta$ controls the Kullback-Leibler divergence (KL) regularization by penalizing models that deviate from the reference policy $\pi_{\theta_{\text{ref}}}$ (which is typically the initial pretrained model). In this work, we utilize a variant of GRPO based on Dynamic sAmpling Policy Optimization (DAPO) (Yu et al., 2025a), specifically adopting its optimization objective while omitting the dynamic sampling component.

## B.3 Rule-based Reward Modeling

Reward modeling is central to reinforcement learning with language models, as it defines the optimization signal guiding the policy $\pi_\theta$. Learned neural reward models are commonly employed to approximate human preferences or task-specific goals. However, they often suffer from distributional mismatch, reward hacking, and spurious correlations (Gao et al., 2023; Weng, 2024; Everitt et al., 2021). These effects arise when the model exploits imperfections in the reward predictor, leading to high-reward outputs that do not correspond to true task success.

An alternative is to design rule-based reward models that define deterministic mappings from model outputs to scalar reward values based on explicit criteria. In the context of coding, for instance, a reward function

$R : \boldsymbol{y} \mapsto [0, 1]$ can be constructed by executing the generated code $\boldsymbol{y}$ against a test suite and returning the fraction of passed unit tests. Such rule-based models directly encode correctness and task satisfaction, avoiding pathologies introduced by learned approximators.

Formally, let $\boldsymbol{x} \sim \mathcal{X}$ denote the input (e.g., a natural language instruction), and $\boldsymbol{y} \sim \pi_{\theta_{\text{old}}}(\cdot \mid \boldsymbol{x})$ a candidate response. The reward function $R(\boldsymbol{x}, \boldsymbol{y}) \in \mathbb{R}$ is defined deterministically via evaluation procedures specified a priori. These functions are task-dependent and vary across application domains. The resulting reward is used to construct advantage estimates, as in GRPO.

A well-known limitation of rule-based reward models is the sparsity of the reward signal. In many structured tasks, the reward $R(\boldsymbol{x}, \boldsymbol{y})$ may remain zero across most model outputs and attain nonzero values only when the generation exactly satisfies task constraints. This sparsity complicates credit assignment during training and may impair exploration in RL-based optimization. Techniques such as reward shaping, curriculum learning, or relaxed matching criteria are sometimes employed to mitigate this issue (Ng, 1999; Sutton et al., 1998; Gao et al., 2023; Narvekar et al., 2020). Nonetheless, provided that the policy starts from a sufficiently strong pretrained model, this approach has been successfully adopted in multiple recent frameworks across general and specialized RLHF pipelines (DeepSeek-AI et al., 2025; Team et al., 2025; Yu et al., 2025b), and has been particularly effective in settings where ground truth verification criteria exist, such as program synthesis (Le et al., 2022; Gehring et al., 2024; Chen et al., 2023).

## C   Training Details and Hyperparameters for RLVR

In this section, we detail the experimental setup for RLVR. We employ the `Qwen3` model family, integrated with the Open-R1 pipeline (HuggingFace, 2025; von Werra et al., 2020) and the DAPO (Yu et al., 2025a) algorithm.

**Hyperparameter Configuration.** Numerical stability was maintained using FlashAttention-2 (Dao et al., 2022), vLLM (Kwon et al., 2023) for efficient inference during RL rollouts and masking out from the loss the truncated completions to avoid unnecessary length rewards based on incomplete generations. In addition, to deal with the training-inference mismatch (Liu et al., 2025a; He & Lab, 2025), the importance sampling is clipped to 2.0 (*vllm_importance_sampling_cap* = 2). The specific hyperparameter configuration is summarized in Table 5.

Table 5: Hyperparameters for RLVR training on the BIRD dataset.

| Hyperparameter | Value |
| --- | --- |
| Optimizer | AdamW |
| Learning Rate | $1 \times 10^{-6}$ |
| Global Batch Size | 256 |
| Micro-batch Size | 64 |
| Rollouts per Batch | 16 |
| Epochs | 1 |
| Max Generation Length | 4096 tokens |
| Temperature ($\tau$) | 0.6 |
| Top-$p$ | 0.95 |
| Repetition Penalty | 1.1 |
| Hardware | 8× NVIDIA H100 (80GB) |
| Software Framework | Open-R1, vLLM, FlashAttention |
| Dtype | bfloat16 |
| mask_truncated_completions | True |

**Prompt Template.** The model is trained with a multi-block prompt structure ( Fig. 7). A `system` message primes the model to reason privately between `<reasoning>` ... `</reasoning>` tags and to surface only executable SQL within `<answer>` tags. The `user` turn provides comprehensive context, including the

---

**RLVR training prompt.**

```
System
You are a data science expert that provides well-reasoned and detailed responses.
Your task is to understand the schema and generate a
valid SQL query to answer the question.
You first think about the reasoning process as an internal monologue
and then provide the user with the answer.
Respond in the following format:
<reasoning>
    ...
</reasoning>
<answer>
    ...
</answer>

User
Answer the following question with the SQL code.
Use the piece of evidence and base your answer on the database schema.
Given the question, the evidence and the database schema,
return in the <answer> tags only the SQL script that addresses the question.

Database Engine:
SQLite

Question:
{ question }

Evidence:
{ evidence }

Database Schema:
{ schema }
```

Figure 7: Prompt used for RLVR training and to evaluate the respective trained model. {question}, {evidence}, and {schema} are placeholders for the actual question, evidence, and database schema, respectively.

```
<reasoning>

To find out how many roles Julia Roberts played in the series, we need to look at the relevant tables in the database
    schema.

From the schema:

— The "Person" table contains information about people, including their IDs and names.
— The "Credit" table links episodes to people via person_id and role, indicating which roles they had in specific episodes
    .
— The "Episode" table has information about each episode, including the series it belongs to.

So, the plan is:

1. Find the person_id for Julia Roberts from the Person table.
2. Join this with the Credit table to get all the roles she has in different episodes.
3. Count those roles.

The SQL query would be:

SELECT COUNT(*) AS num_roles
FROM Credit c
JOIN Person p ON c.person_id = p.person_id
WHERE p.name = 'Julia Roberts';
</reasoning>

<answer>
SELECT COUNT(*) AS num_roles
FROM Credit c
JOIN Person p ON c.person_id = p.person_id
WHERE p.name = 'Julia Roberts';
</answer>",
```

Figure 8: Reasoning example where `Qwen3-4B` succeeded during training. Question *"How many roles did Julia Roberts play in the series?"*, target query: `"SELECT COUNT(T1.role)FROM Credit AS T1 INNER JOIN Person AS T2 ON T2.person_id = T1.person_id WHERE T2.name = 'Julia Roberts'"`.

natural-language question, evidence, and database schema, ensuring that all necessary tokens for generation are present in a single prompt. For SFT initialization, the model is teacher-forced on both latent reasoning and the final SQL output, establishing a structural contract that simplifies reward parsing during subsequent RL stages. In Fig. 8 and Fig. 9, we show examples of reasoning traces generated by `Think2SQL-4B` on the BIRD training set, where the model succeeds and fails, respectively.

## D  Rewards for RLVR

In RLVR, reward signals are crucial for guiding the model's learning process (Gao et al., 2023; Weng, 2024; Everitt et al., 2021). Execution accuracy, the primary reward for Text-to-SQL, measures the correctness of generated SQL by comparing it to the ground truth. However, its binary nature poses challenges for RL optimization, particularly for smaller LLMs, because rewards typically remain zero unless the SQL is exactly correct. To address this limitation, we integrate QATCH (Papicchio et al., 2023; 2025b), an advanced benchmarking framework designed for the automated evaluation of Text2SQL tasks. For the purposes of this study, we employed three primary QATCH metrics: *Cell Precision*, *Cell Recall*, and *Tuple Cardinality*.

To encourage the model's reasoning process, we introduce the *Format reward* that evaluates the appropriate use of reasoning tags (DeepSeek-AI et al., 2025).

<reasoning>
To find the trips that started at "Mountain View City Hall" and ended on a rainy day, we need to look into the "trip" table and the "weather" table.

From the schema:
— The "trip" table has a column called "start_station_name", which contains the names of the starting stations.
— The "trip" table also has a column called "end_date".
— The "weather" table has a column called "events", which can be "rain", "Rain", etc., indicating whether it rained on a given date.
— The "weather" table has a column called "date", which represents the date of the weather event.
— The "trip" table's "end_date" should match the "date" from the "weather" table for the same day, but note that the "end_date" in the "trip" table seems to have time information (e.g., "8/29/2013 9:11"), while the "date" in the "weather" table is just the date without time (e.g., "8/29/2013").

However, since the question says "ended on a rainy day", it likely means that the day of the trip's end was a rainy day. Therefore, we need to check if the "end_date" corresponds to a date in the "weather" table where "events" is "rain".

So, the plan is:

1. Find all trips where "start_station_name" is "Mountain View City Hall".
2. For each such trip, check if the "end_date" (without time) matches a "date" in the "weather" table where "events" is "rain".

Since the "end_date" in the "trip" table includes a time component, we need to extract the date part from it to compare with the "date" field in the "weather" table.

In SQLite, we can use the `SUBSTR` function to extract the date part from "end_date". For example, if "end_date" is "8/29/2013 9:11", then `SUBSTR(end_date, 1, 10)` would give "8/29/2013".

Then, we can join the "trip" table with the "weather" table on this condition.

Therefore, the SQL query would be:

SELECT DISTINCT t.id
FROM trip t
JOIN weather w ON SUBSTR(t.end_date, 1, 10) = w.date
WHERE t.start_station_name = 'Mountain View City Hall'
  AND w.events = 'rain';
</reasoning>

<answer>
SELECT DISTINCT t.id
FROM trip t
JOIN weather w ON SUBSTR(t.end_date, 1, 10) = w.date
WHERE t.start_station_name = 'Mountain View City Hall'
  AND w.events = 'rain';
</answer>",

Figure 9: Reasoning example where `Qwen3-4B` fail during training. Question *"Which were the trips that started at Mountain View City Hall and ended on a rainy day?"*, target query: "`SELECT` T1.id `FROM` trip `AS` T1 `INNER JOIN` weather `AS` T2 `WHERE` T2.events = `'Rain'AND` T1.start_station_name = `'Mountain View City Hall'`".

Let $\mathcal{T}$ and $\mathcal{T}_{\mathrm{pred}}$ denote the execution results of the target SQL query and the predicted SQL query, respectively, each represented as a set of tuples, where each tuple comprises a set of cell values. The reward signals utilized in this study are outlined below:

**Execution Accuracy (EX).** EX (Yu et al., 2018; Li et al., 2024b) evaluates whether the execution of the target SQL query matches the execution of the predicted SQL query. It is defined as:

$$R_{\text{EX}} = \begin{cases} 1 & \text{if } \mathcal{T} = \mathcal{T}_{\text{pred}} \\ 0 & \text{otherwise} \end{cases}, \quad R_{\text{EX}} \in \{0, 1\}$$

This metric provides a binary reward, assigning a full score only when the two execution results match exactly, row by row. While straightforward and reliable, execution accuracy does not account for partially correct results, which can hinder the learning process in RL.

**Cell Precision (CP).** CP is the fraction of table cells in $\mathcal{T}_{\text{pred}}$ that are in the target $\mathcal{T}$. The higher the score, the more predicted cells are in the target.

$$R_{\text{CP}} = \frac{|\{cells \mid cells \in \mathcal{T} \cap \mathcal{T}_{\text{pred}}\}|}{|\{cells \mid cells \in \mathcal{T}_{\text{pred}}\}|} \in [0, 1]$$

This metric allows partial credit when the predicted SQL query execution includes some requested cells but also contains incorrect ones. Considering the target query `SELECT` Name `FROM` Player; and the predicted query `SELECT` Name, Surname `FROM` Player; In this case, CP is 0.5 because the predicted SQL query execution contains the correct cells from the Name column and incorrect cells from the Surname column. However, it does not consider whether all the requested cells in $\mathcal{T}$ are present in the SQL query - measured by *Cell Recall*. It is worth noticing that when EX is 1 also CP is 1.

**Cell Recall (CR).** CR is the fraction of table cells in $\mathcal{T}$ that are present in $\mathcal{T}_{\text{pred}}$. The higher the score, the more target cells are included in the prediction.

$$R_{\text{CR}} = \frac{|\{cells \mid cells \in \mathcal{T} \cap \mathcal{T}_{\text{pred}}\}|}{|\{cells \mid cells \in \mathcal{T}\}|} \in [0, 1]$$

This metric allows a partial reward when the predicted SQL query does not contain all the requested cells in the target query. Considering the target query `SELECT` Name, Surname `FROM` Player; and the predicted query `SELECT` Name, ~~Surname~~ `FROM` Player; then CP is 1 but CR is 0.5 because the predicted SQL query execution contains the correct cells from the column Name but not from Surname. It is worth noting that when EX = 1, CR = 1 as well.

**Tuple Cardinality (TC).** TC is defined as the ratio between the number of tuples in $\mathcal{T}_{\text{pred}}$ and the number of tuples in $\mathcal{T}$. The min function is used to ensure TC $\in [0, 1]$. TC captures output cardinality only, ignoring schema and cell values. Thus, it should be considered alongside CP and CR for a fuller view of model performance. The TC reward is defined as:

$$R_{\text{TC}} = \min\left(\frac{|\mathcal{T}|}{|\mathcal{T}_{\text{pred}}|}, \frac{|\mathcal{T}_{\text{pred}}|}{|\mathcal{T}|}\right) \in [0, 1]$$

This metric is necessary because CP and CR are computed from the intersection of cell values, which may obscure differences in output size when the number of cells is not critical. For example, consider the target query `SELECT DISTINCT` Name `FROM` Player; and the predicted query `SELECT` ~~DISTINCT~~ Name `FROM` Player;. In this case, both CP and CR equal 1 due to identical cell values, yet the cardinality of the cells is different. Thus, the TC metric is essential for capturing this discrepancy.

**Format Reward (FR).** The FR incentivizes the model to adhere to a predefined output structure, such as the use of `<reasoning>` and `<answer>` tags.

$$R_{\text{FR}} = \mathbb{1}_{\{\text{format matches}\}}(\pi_{\boldsymbol{\theta}}(\boldsymbol{x})) \tag{8}$$

This is a sparse reward that activates only when both the opening and closing tags for reasoning and answers are correctly positioned. The reward value is 1 if the tags are correctly formatted; otherwise, it is 0.

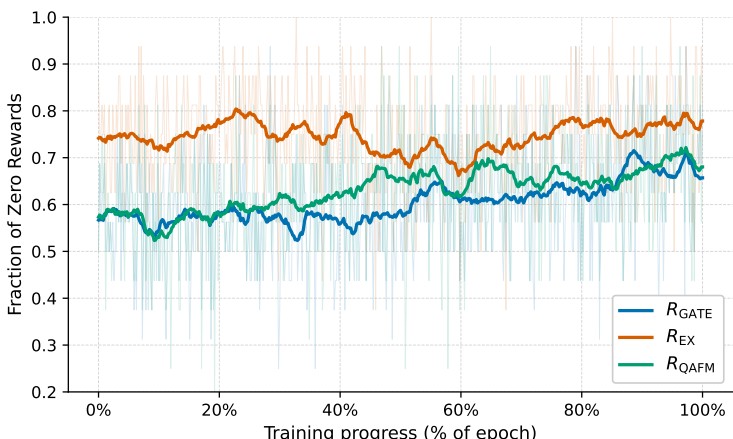

Figure 10: **Fraction of zero-reward groups during training.** The plot shows the fraction of groups with zero reward across training iterations for `Qwen3-4B` across different reward configurations.

## E  Full Results Across Different Configurations

This section reports the complete results presented in the main paper, including all reward configurations and evaluation metrics across all datasets. The results are presented in comprehensive tables that enable a detailed comparison of model performance across various training conditions.

**Results Different Rewards and Training configurations** Table 6 reports the detailed results for different reward configurations and training settings, including execution accuracy ($R_{\mathrm{EXFM}}$) and QATCH metrics ($R_{\mathrm{QAFM}}$ and $R_{\mathrm{GATE}}$) across all datasets, illustrating the interplay between reward design, advantage estimation, and model scale. The data reveals a clear divergence in behavior based on parameter count. For smaller 4B architectures, dense reward signals guide the model to better performance. Conversely, larger 8B+ models show a higher propensity to exploit local heuristics within dense rewards, leading to suboptimal performance as evidenced by lower $R_{\mathrm{GATE}}$ scores. However, the 14B model's superior capacity partially offsets this risk, narrowing the performance gap between reward strategies.

**Full Results on Cold Start** Table 7 presents detailed results for different cold-start configurations, highlighting the impact of initial model states on performance across various datasets and metrics.

**Full Results on Pareto Frontier** Table 8 presents detailed results for different dataset sampling configurations, illustrating trade-offs among training data size, model performance, and generalization across benchmarks.

## F  Additional Analysis on Training plots

This section reports additional training plots for different reward configurations and training settings, providing insights into the learning dynamics and convergence behavior of models under various conditions.

**Zero-variance groups for `Qwen3-4B`.** The Fig. 10 reports the fraction of zero-reward groups during training for the 4B model across different reward configurations introduced in the main paper. $R_{\mathrm{QAFM}}$ and $R_{\mathrm{GATE}}$ reward configurations are dense rewards instead $R_{\mathrm{EXFM}}$ is a sparse reward. The plot clearly shows that the fraction of zero-reward groups is significantly higher for $R_{\mathrm{EXFM}}$ compared to $R_{\mathrm{QAFM}}$ and $R_{\mathrm{GATE}}$, which is expected given the binary nature of the execution accuracy reward. The dense reward configurations ($R_{\mathrm{QAFM}}$ and $R_{\mathrm{GATE}}$) provide more informative feedback during training, resulting in a lower fraction of zero-reward groups facilitating better learning dynamics for the 4B model.

**The Initialization Bias for `Qwen3-4B`.** Fig. 11 reports the training dynamics under different cold-start configurations for the `Qwen3-4B`. The left panel displays the average token-level entropy of the output

Table 6: **Scaling Laws and Reward Normalization Dynamics in RLVR.** Comparative performance (EX) of RLVR-trained `Qwen3` models (4B, 8B, and 14B) under different advantage normalization strategies (*No Scale*, *Group Scaling*, and *Batch Scaling*) and reward formulations ($R_{\mathrm{QAFM}}$, $R_{\mathrm{GATE}}$, and $R_{\mathrm{EXFM}}$). Models are trained on BIRD-TRAIN and evaluated on BIRD-DEV, SPIDER variants and EHRSQL. *Base* and *Thinking* represent `Qwen-4B` in its standard and reasoning configurations. *Weig. AVG.* is the weighted average performance across all dataset by dataset size. *Weig. AVG. (OOD)* is the weighted average performance across Out-Of-Distribution (OOD) benchmarks (SPIDER variants and EHRSQL).

| LLM | BIRD (dev) #1,530 | Spider test #2,147 | Spider-DK #1,034 | Spider-Syn #535 | Spider-Realistic #508 | EHRSQL #1,008 | Weig. AVG. | Weig. AVG. (OOD) |
|---|---|---|---|---|---|---|---|---|
| **Study Scale Qwen4B** | | | | | | | | |
| Base | $49.0_{\pm 0.5}$ | $78.6_{\pm 0.6}$ | $68.3_{\pm 1.5}$ | $68.0_{\pm 1.7}$ | $76.5_{\pm 0.9}$ | $29.0_{\pm 1.9}$ | 61.9 | 65.7 |
| Thinking | $52.0_{\pm 0.5}$ | $80.2_{\pm 0.2}$ | $70.7_{\pm 0.5}$ | $71.4_{\pm 1.0}$ | $78.8_{\pm 1.0}$ | $30.3_{\pm 0.8}$ | **64.1** | **67.7** |
| **Scale Group** | | | | | | | | |
| $R_{\mathrm{EXFM}}$ | $54.5_{\pm 0.7}$ | $81.1_{\pm 0.1}$ | $70.4_{\pm 1.3}$ | $70.5_{\pm 0.1}$ | $77.9_{\pm 0.6}$ | $28.1_{\pm 1.4}$ | 64.5 | 67.4 |
| $R_{\mathrm{QAFM}}$ | $55.8_{\pm 0.1}$ | $\mathbf{85.5_{\pm 0.4}}$ | $76.0_{\pm 1.1}$ | $\mathbf{75.5_{\pm 0.7}}$ | $\mathbf{82.5_{\pm 0.7}}$ | $32.8_{\pm 0.4}$ | 68.5 | **72.2** |
| $R_{\mathrm{GATE}}$ | $\mathbf{59.6_{\pm 0.3}}$ | $83.7_{\pm 0.2}$ | $\mathbf{76.0_{\pm 0.3}}$ | $75.0_{\pm 0.5}$ | $81.2_{\pm 0.1}$ | $\mathbf{33.5_{\pm 0.3}}$ | **68.7** | 71.4 |
| **Scale Batch** | | | | | | | | |
| $R_{\mathrm{EXFM}}$ | $\mathbf{58.2_{\pm 0.7}}$ | $83.7_{\pm 0.6}$ | $\mathbf{75.7_{\pm 0.3}}$ | $72.9_{\pm 0.3}$ | $81.5_{\pm 0.6}$ | $32.2_{\pm 1.0}$ | **68.0** | 70.9 |
| $R_{\mathrm{QAFM}}$ | $54.9_{\pm 1.2}$ | $81.6_{\pm 0.3}$ | $73.1_{\pm 1.9}$ | $73.4_{\pm 0.8}$ | $81.0_{\pm 0.6}$ | $32.9_{\pm 0.6}$ | 66.3 | 69.6 |
| $R_{\mathrm{GATE}}$ | $57.8_{\pm 0.4}$ | $82.7_{\pm 0.4}$ | $75.5_{\pm 0.5}$ | $\mathbf{74.8_{\pm 0.3}}$ | $81.9_{\pm 0.6}$ | $\mathbf{33.9_{\pm 0.7}}$ | **68.0** | **71.0** |
| **No Scale** | | | | | | | | |
| $R_{\mathrm{EXFM}}$ | $57.2_{\pm 0.9}$ | $\mathbf{85.0_{\pm 0.4}}$ | $74.5_{\pm 0.5}$ | $73.7_{\pm 0.4}$ | $80.4_{\pm 1.3}$ | $30.3_{\pm 1.0}$ | 67.7 | 70.8 |
| $R_{\mathrm{QAFM}}$ | $\mathbf{58.0_{\pm 0.6}}$ | $84.3_{\pm 0.6}$ | $73.8_{\pm 0.6}$ | $73.4_{\pm 0.5}$ | $\mathbf{81.5_{\pm 0.5}}$ | $\mathbf{32.5_{\pm 0.6}}$ | **67.9** | 70.9 |
| $R_{\mathrm{GATE}}$ | $56.2_{\pm 0.3}$ | $84.1_{\pm 0.3}$ | $\mathbf{75.7_{\pm 0.4}}$ | $\mathbf{74.5_{\pm 0.3}}$ | $81.2_{\pm 1.0}$ | $31.2_{\pm 0.7}$ | 67.6 | **71.0** |
| **Study Scale Qwen-8B** | | | | | | | | |
| Base | $52.0_{\pm 0.6}$ | $81.9_{\pm 0.2}$ | $71.7_{\pm 1.5}$ | $71.4_{\pm 0.8}$ | $79.4_{\pm 0.2}$ | $32.4_{\pm 0.4}$ | **65.2** | **69.0** |
| Thinking | $53.5_{\pm 0.1}$ | $78.5_{\pm 0.3}$ | $69.9_{\pm 0.7}$ | $70.3_{\pm 0.2}$ | $78.3_{\pm 0.8}$ | $30.1_{\pm 0.6}$ | 63.6 | 66.6 |
| **Scale Group** | | | | | | | | |
| $R_{\mathrm{EXFM}}$ | $55.5_{\pm 0.2}$ | $84.0_{\pm 0.2}$ | $\mathbf{78.2_{\pm 0.5}}$ | $74.0_{\pm 0.7}$ | $81.3_{\pm 0.1}$ | $29.3_{\pm 0.5}$ | 67.5 | 71.0 |
| $R_{\mathrm{QAFM}}$ | $57.6_{\pm 0.3}$ | $\mathbf{84.3_{\pm 0.6}}$ | $75.9_{\pm 0.4}$ | $75.4_{\pm 0.6}$ | $80.1_{\pm 0.8}$ | $35.5_{\pm 0.5}$ | **68.7** | **71.9** |
| $R_{\mathrm{GATE}}$ | $\mathbf{57.8_{\pm 0.3}}$ | $82.4_{\pm 0.3}$ | $75.8_{\pm 0.8}$ | $\mathbf{75.6_{\pm 0.5}}$ | $\mathbf{81.4_{\pm 1.4}}$ | $\mathbf{38.5_{\pm 0.6}}$ | **68.7** | 71.8 |
| **Scale Batch** | | | | | | | | |
| $R_{\mathrm{EXFM}}$ | $\mathbf{58.8_{\pm 0.4}}$ | $83.7_{\pm 0.5}$ | $\mathbf{75.9_{\pm 0.7}}$ | $\mathbf{74.9_{\pm 0.6}}$ | $80.9_{\pm 0.6}$ | $32.9_{\pm 0.4}$ | 68.4 | 71.2 |
| $R_{\mathrm{QAFM}}$ | $56.0_{\pm 0.4}$ | $\mathbf{84.7_{\pm 0.5}}$ | $75.4_{\pm 0.1}$ | $74.3_{\pm 0.5}$ | $\mathbf{81.5_{\pm 0.8}}$ | $\mathbf{36.3_{\pm 1.1}}$ | **68.5** | **72.2** |
| $R_{\mathrm{GATE}}$ | $56.8_{\pm 1.0}$ | $82.0_{\pm 0.2}$ | $75.0_{\pm 1.1}$ | $73.5_{\pm 0.5}$ | $79.6_{\pm 0.5}$ | $29.5_{\pm 0.4}$ | 66.5 | 69.4 |
| **No Scale** | | | | | | | | |
| $R_{\mathrm{EXFM}}$ | $57.0_{\pm 0.8}$ | $82.0_{\pm 0.5}$ | $75.2_{\pm 1.4}$ | $73.7_{\pm 0.6}$ | $78.5_{\pm 0.0}$ | $\mathbf{31.5_{\pm 0.4}}$ | 66.9 | 69.7 |
| $R_{\mathrm{QAFM}}$ | $55.3_{\pm 0.6}$ | $\mathbf{83.8_{\pm 0.1}}$ | $\mathbf{76.5_{\pm 0.1}}$ | $\mathbf{76.5_{\pm 0.8}}$ | $\mathbf{81.8_{\pm 1.0}}$ | $30.9_{\pm 0.6}$ | **67.6** | **71.2** |
| $R_{\mathrm{GATE}}$ | $55.2_{\pm 0.7}$ | $83.5_{\pm 0.2}$ | $73.8_{\pm 0.4}$ | $72.8_{\pm 0.5}$ | $79.6_{\pm 0.9}$ | $31.2_{\pm 0.6}$ | 66.7 | 70.0 |
| **Study Scale Qwen-14B** | | | | | | | | |
| Base | $55.2_{\pm 0.4}$ | $83.9_{\pm 0.4}$ | $74.0_{\pm 1.4}$ | $73.2_{\pm 0.5}$ | $83.1_{\pm 1.6}$ | $32.2_{\pm 1.0}$ | **67.3** | **70.8** |
| Thinking | $55.3_{\pm 0.3}$ | $81.0_{\pm 0.4}$ | $73.4_{\pm 0.4}$ | $73.3_{\pm 0.3}$ | $80.6_{\pm 0.3}$ | $33.5_{\pm 0.8}$ | 66.3 | 69.5 |
| **Scale Group** | | | | | | | | |
| $R_{\mathrm{EXFM}}$ | $\mathbf{61.3_{\pm 0.8}}$ | $\mathbf{85.6_{\pm 0.4}}$ | $77.8_{\pm 0.9}$ | $\mathbf{78.6_{\pm 0.1}}$ | $\mathbf{81.8_{\pm 0.6}}$ | $38.9_{\pm 0.3}$ | **71.3** | **74.0** |
| $R_{\mathrm{QAFM}}$ | $61.1_{\pm 0.1}$ | $84.5_{\pm 0.2}$ | $76.2_{\pm 0.6}$ | $75.6_{\pm 0.6}$ | $79.4_{\pm 0.5}$ | $\mathbf{39.8_{\pm 0.6}}$ | 70.2 | 72.8 |
| $R_{\mathrm{GATE}}$ | $58.8_{\pm 0.1}$ | $83.2_{\pm 0.2}$ | $\mathbf{77.9_{\pm 0.5}}$ | $75.4_{\pm 0.4}$ | $80.2_{\pm 0.3}$ | $36.0_{\pm 1.4}$ | 69.0 | 72.0 |
| **Scale Batch** | | | | | | | | |
| $R_{\mathrm{EXFM}}$ | $61.4_{\pm 0.2}$ | $84.3_{\pm 0.4}$ | $75.7_{\pm 0.4}$ | $75.6_{\pm 0.4}$ | $80.6_{\pm 0.3}$ | $35.5_{\pm 1.4}$ | **69.6** | 71.9 |
| $R_{\mathrm{QAFM}}$ | $\mathbf{61.5_{\pm 0.2}}$ | $\mathbf{84.6_{\pm 0.1}}$ | $\mathbf{77.9_{\pm 0.7}}$ | $76.1_{\pm 0.8}$ | $\mathbf{81.4_{\pm 0.5}}$ | $32.2_{\pm 0.4}$ | **69.6** | **72.0** |
| $R_{\mathrm{GATE}}$ | $61.0_{\pm 0.4}$ | $83.6_{\pm 0.1}$ | $77.1_{\pm 1.0}$ | $\mathbf{76.5_{\pm 0.9}}$ | $79.6_{\pm 0.7}$ | $\mathbf{35.5_{\pm 0.5}}$ | 69.5 | 71.9 |
| **No Scale** | | | | | | | | |
| $R_{\mathrm{EXFM}}$ | $\mathbf{60.9_{\pm 0.4}}$ | $84.8_{\pm 0.3}$ | $76.5_{\pm 0.6}$ | $\mathbf{76.7_{\pm 0.8}}$ | $81.6_{\pm 0.4}$ | $\mathbf{38.0_{\pm 0.9}}$ | **70.3** | **73.0** |
| $R_{\mathrm{QAFM}}$ | $59.4_{\pm 0.1}$ | $84.0_{\pm 0.2}$ | $77.3_{\pm 0.6}$ | $76.0_{\pm 0.6}$ | $80.4_{\pm 0.3}$ | $33.7_{\pm 0.9}$ | 69.0 | 71.8 |
| $R_{\mathrm{GATE}}$ | $59.9_{\pm 0.2}$ | $\mathbf{85.1_{\pm 0.3}}$ | $\mathbf{77.7_{\pm 1.1}}$ | $75.8_{\pm 0.7}$ | $\mathbf{82.1_{\pm 0.9}}$ | $31.9_{\pm 0.2}$ | 69.4 | 72.1 |

Table 7: **Cold-Start impact on RLVR Performance.** Performance comparison of different training strategies for `Qwen-4B` trained on BIRD-TRAIN. All RLVR experiments use the $R_{\mathrm{GATE}}$ configuration with group scaling. *Base* and *Thinking* denote `Qwen-4B` in Base and *Thinking* model configuration; $SFT_{NT}$ denotes the SFT without reasoning traces. $SFT_D$ denotes the SFT on our constructed datasets $\mathcal{D}_{SFT}$. $RL_{SFT_D}$ denotes RLVR starting from the SFT-trained model $SFT_D$. $RL_{\mathrm{Think}}$ denotes RLVR starting from the *Thinking* configuration.

| LLM | BIRD (dev) #1,530 | Spider test #2,147 | Spider-DK #1,034 | Spider-Syn #535 | Spider-Realistic #508 | EHRSQL #1,008 |
|---|---|---|---|---|---|---|
| *Base* | $49.0_{\pm 0.5}$ | $78.6_{\pm 0.6}$ | $68.3_{\pm 1.5}$ | $68.0_{\pm 1.7}$ | $76.5_{\pm 0.9}$ | $29.0_{\pm 1.9}$ |
| *Thinking* | $52.0_{\pm 0.5}$ | $80.2_{\pm 0.2}$ | $70.7_{\pm 0.5}$ | $71.4_{\pm 1.0}$ | $78.8_{\pm 1.0}$ | $30.3_{\pm 0.8}$ |
| $SFT_{NT}$ | $46.5_{\pm 0.5}$ | $59.9_{\pm 0.3}$ | $56.4_{\pm 1.1}$ | $48.1_{\pm 1.0}$ | $56.2_{\pm 0.5}$ | $15.7_{\pm 0.5}$ |
| $SFT_D$ | $55.5_{\pm 0.2}$ | $84.2_{\pm 0.0}$ | $78.2_{\pm 0.5}$ | $74.0_{\pm 0.7}$ | $81.3_{\pm 0.1}$ | $21.1_{\pm 1.3}$ |
| $R_{\mathrm{GATE}}$ | $59.6_{\pm 0.3}$ | $83.7_{\pm 0.2}$ | $76.0_{\pm 0.3}$ | $75.0_{\pm 0.5}$ | $81.2_{\pm 0.1}$ | $33.5_{\pm 0.3}$ |
| $RL_{SFT_D}$ | $59.4_{\pm 0.2}$ | $82.8_{\pm 0.2}$ | $73.4_{\pm 1.2}$ | $74.7_{\pm 1.1}$ | $78.7_{\pm 1.0}$ | $26.2_{\pm 0.4}$ |
| $RL_{\mathrm{Think}}$ | $49.2_{\pm 0.3}$ | $76.4_{\pm 0.3}$ | $69.4_{\pm 0.8}$ | $68.2_{\pm 0.4}$ | $73.4_{\pm 0.5}$ | $24.0_{\pm 0.7}$ |

Table 8: **Model performance for different dataset subsampling ratios.** Results of RLVR-trained `Qwen-4B` on BIRD-TRAIN and evaluated on BIRD-DEV, SPIDER variants and EHRSQL with different subsampling ratios ($RL_{1:n}$). *Base* and *Thinking* refer to `Qwen-4B` in *Base* and *Thinking* mode, respectively.

| LLM | BIRD (dev) #1,530 | Spider test #2,147 | Spider-DK #1,034 | Spider-Syn #535 | Spider-Realistic #508 | EHRSQL #1,008 |
|---|---|---|---|---|---|---|
| *Base* | $49.0_{\pm 0.5}$ | $78.6_{\pm 0.6}$ | $68.3_{\pm 1.5}$ | $68.0_{\pm 1.7}$ | $76.5_{\pm 0.9}$ | $29.0_{\pm 1.9}$ |
| *Thinking* | $52.0_{\pm 0.5}$ | $80.2_{\pm 0.2}$ | $70.7_{\pm 0.5}$ | $71.4_{\pm 1.0}$ | $78.8_{\pm 1.0}$ | $30.3_{\pm 0.8}$ |
| $RL_{1:1}$ | $\mathbf{59.6_{\pm 0.3}}$ | $83.7_{\pm 0.2}$ | $\mathbf{76.0_{\pm 0.3}}$ | $75.0_{\pm 0.5}$ | $81.2_{\pm 0.1}$ | $33.5_{\pm 0.3}$ |
| $RL_{1:2}$ | $58.4_{\pm 0.4}$ | $\mathbf{84.5_{\pm 0.5}}$ | $74.6_{\pm 0.6}$ | $73.8_{\pm 0.8}$ | $\mathbf{81.7_{\pm 0.3}}$ | $\mathbf{37.9_{\pm 1.2}}$ |
| $RL_{1:4}$ | $57.6_{\pm 0.3}$ | $83.0_{\pm 0.3}$ | $74.8_{\pm 0.7}$ | $74.6_{\pm 0.3}$ | $80.8_{\pm 1.2}$ | $37.4_{\pm 0.1}$ |
| $RL_{1:8}$ | $55.8_{\pm 0.6}$ | $83.2_{\pm 0.3}$ | $73.2_{\pm 0.8}$ | $74.7_{\pm 0.6}$ | $80.9_{\pm 1.0}$ | $34.4_{\pm 0.2}$ |
| $RL_{1:16}$ | $52.2_{\pm 0.3}$ | $81.6_{\pm 0.1}$ | $73.4_{\pm 1.2}$ | $71.9_{\pm 0.4}$ | $77.6_{\pm 0.8}$ | $29.9_{\pm 0.7}$ |

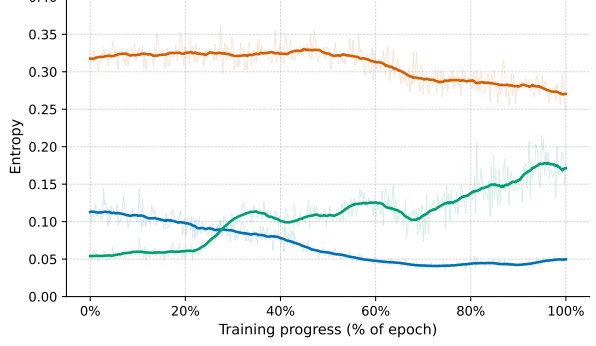
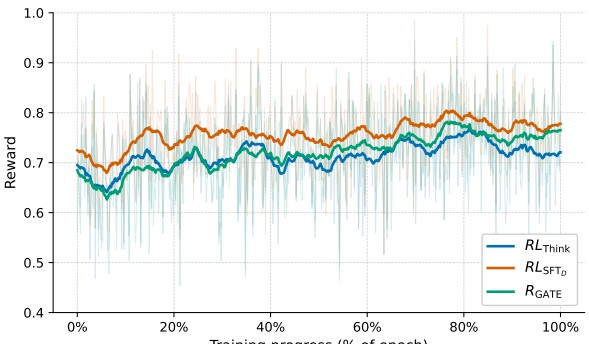

Figure 11: **Training dynamics for different cold-start configurations for `Qwen3-4B`.** The left plot shows the average token-level entropy of the model's output distribution across training iterations for different cold-start configurations. The right plot shows the average reward obtained by the model across training iterations for different cold-start configurations. All RLVR experiments use the $R_{\mathrm{GATE}}$ configuration with group scaling. $RL_{SFT_D}$ denotes RLVR starting from the SFT-trained model $SFT_D$. $RL_{\mathrm{Think}}$ denotes RLVR starting from the *Thinking* `Qwen3-4B` configuration.

distribution over the course of training, while the right panel tracks the corresponding average reward. All RLVR experiments adopt the $R_{\text{GATE}}$ configuration with group scaling. The results provide clear evidence that the choice of initialization introduces a non-trivial bias into the subsequent training dynamics. The model initialized with our curated dataset $SFT_D$ exhibits markedly higher token-level entropy in the early stages of training, attributable to the distributional shift between the supervised fine-tuning data and the model's internal distribution. However, this entropy declines steadily, indicating that the model converges prematurely and fails to explore diverse reasoning trajectories. A complementary phenomenon is observed for $RL_{\text{Think}}$, the variant initialized with the *Thinking* configuration: it begins training with substantially lower entropy, reflecting high initial confidence, yet similarly undergoes entropy collapse, suggesting an inability to discover novel reasoning paths beyond those encoded during initialization.

In contrast, the base `Qwen3-4B` initialization yields a qualitatively different and more favorable training profile. Its token-level entropy increases steadily throughout training, consistent with a model that progressively broadens its exploration of the reasoning space without being anchored to a biased starting distribution. This advantage is corroborated by the reward curves: although the *Qwen3-4B* initialization starts from the lowest reward among the three configurations, it surpasses $RL_{\text{Think}}$ and sustains a consistent upward trajectory, whereas both $SFT_D$ and $RL_{\text{Think}}$ exhibit diminishing reward gains as training progresses.

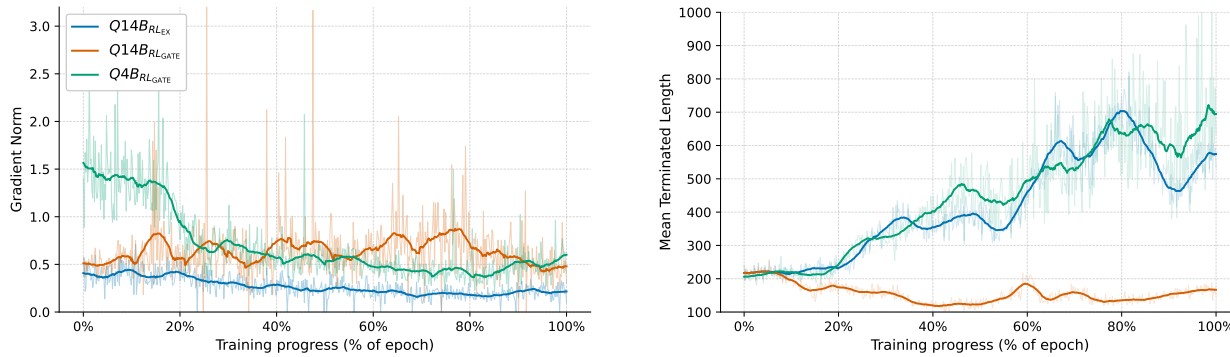

Figure 12: **Gradient Norm and Completion Length Dynamics Across Model Scales.** (Left) Total L2 gradient norm across all trainable parameters over the course of training. (Right) Mean completion length of the $G$ generated outputs per training step. All models use Group Scaling. $Q14B_{RL_{\text{GATE}}}$ and $Q14B_{RL_{\text{EX}}}$ denote `Qwen3-14B` trained with $R_{\text{GATE}}$ and $R_{\text{EXFM}}$, respectively; $Q4B_{RL_{\text{GATE}}}$ denotes `Qwen3-4B` trained with $R_{\text{GATE}}$.

**Reward hacking for larger models.** Fig. 12 reports the training dynamics of the gradient norm and completion length across model scales for different reward configurations. The left panel shows the total L2 gradient norm across all trainable parameters over the course of training, while the right panel tracks the mean completion length of the $G$ generated outputs per training step. The results reveal a clear divergence in training dynamics based on model scale and reward configuration.

The larger `Qwen3-14B` model trained with $R_{\text{GATE}}$ exhibits persistent high-magnitude gradient spikes throughout training, with peaks exceeding $3\times$ the baseline magnitude. These spikes indicate that the policy is making abrupt distributional shifts across batches, consistent with the model oscillating between locally exploitative strategies rather than following a smooth optimization trajectory. This interpretation is corroborated by the right panel: the same configuration shows a stable mean completion length, suggesting that the model has converged on a fixed generation strategy whose reward varies erratically across batches rather than progressively expanding its reasoning capacity. Together, these signals constitute evidence with reward shortcut exploitation: the model exploits sharp, non-robust reward modes that yield high returns on specific batches but fail to generalize.

In contrast, the `Qwen3-4B` model trained with $R_{\text{GATE}}$ displays a qualitatively different and more favorable training profile. Its gradient norm is elevated in the initial phase of training, reflecting the large corrective signal required when the policy is far from the reward-maximizing region, but declines monotonically thereafter,

indicating that each training step builds incrementally on the previous one without contradictory gradient signals. The completion length for this configuration increases monotonically, confirming that the model progressively discovers longer and more elaborate reasoning paths. This divergence supports our earlier finding that smaller models are implicitly regularized by their limited capacity, which prevents them from discovering and systematically exploiting reward shortcuts that larger models can leverage.

Furthermore, the `Qwen3-14B` model trained with $R_{\text{EXFM}}$ follows a trajectory similar to the 4B model, exhibiting a smoothly increasing completion length and a controlled gradient norm. Notably, the gradient norm starts from a lower absolute value, reflecting the fact that the 14B model's stronger prior knowledge places it closer to the reward-maximizing region at initialization. This observation reinforces our central finding from RQ1: sparse reward formulations are better suited for larger models, as they provide a less exploitable optimization landscape that channels model capacity toward genuine task improvement rather than reward shortcuts.

