# OpenReview forum: "Think2SQL: Blueprinting Reward Density and Advantage Scaling for Effective Text-to-SQL Reasoning"
_TMLR — Accepted by TMLR_

### Review · Reviewer_B7KX · 2026-02-24

**Summary Of Contributions:**

This paper is an empirical study on training a reasoning model through Reinforcement Learning with Verifiable Rewards (RLVR) for text-to-SQL problems. The authors provides a refined execution accuracy metric in an extensible python toolkit for better evaluation, curated a dataset by distilling Gemini-3-flash, and train Qwen3 models through Supervised Fine-Tuning (SFT) and DAPO. A dense, execution-guided reward signal is designed and find to be far outperforming binary signals. The experiment results find that 1) a consistent high-fidelity reward signal is essential, with small models benefits more from dense reward with group scaling while larger models requires sparse reward with aggressive batch scaling to prevent overfitting; 2) SFT boosts performance immediately but limits the ceiling of RLVR, and 3) RLVR with moderate data subsampling enhances generalization and outperforms SFT.

**Key Strengths**

1. The paper is well-written and easy to follow; the research questions proposed are clearly listed and answered in corresponding sections, and each components (reward, evaluation, training etc.) are clearly stated. I particularly appreciate that the authors provide a summary for each research questions in the experiment section.

2. There are multiple contributions provided by this paper: a reward recipe, a detailed result and analysis for training, a ready-to-use family of model, a high-quality dataset and an evaluation kit. All of these can be valuable to future works in the community.

3. The experiment and discussion are solid and clearly showing the advantage of the authors' proposed training techniques, as well as proving the conclusions that answers the proposed research questions.

**Key Weaknesses**

1. Some of the findings in the empirical study are already known to the RLVR community (see TMLR's audiences' interest section).

2. The related work section is insufficient (see requested change section).

**Audience:**

Yes

**Audience Explanation:**

This paper is an empirical study on RLVR with the text-to-SQL task. This topic should be interesting to people who work in the text-to-SQL area, not only for the empirical study, but also for the model, reward recipe, dataset and evaluation package provided by the work. For people who work in the general RLVR, some findings are less interesting (e.g. RLVR is more robust than SFT, which is well-known in the community [1, 2, 5]; SFT can lower the upper bound of performance [3, 4, 5, 6] - I do not mean to ask the authors to compare to some of the newest papers, but these paper points to previous works of the similar topic). Nevertheless, there are still some findings that are valuable (e.g. group scaling vs. batch scaling, large models requires sparser signals).

**References**

[1] https://arxiv.org/abs/2505.19789

[2] https://arxiv.org/html/2602.10815

[3] https://arxiv.org/abs/2504.11468

[4] https://arxiv.org/pdf/2602.01058

[5] https://arxiv.org/pdf/2506.01939

[6] https://arxiv.org/pdf/2510.01624

**Broader Impact Concerns:**

While the authors do not discuss broader impact, I do not see any significant broader impact concern that makes the statement a must.

**Claims And Evidence:**

Yes

**Claims Explanation:**

The main contributions of the works are: 1. the reward function design, 2. a comprehensive analysis that shows the optimal training, computation resource and reward strategy varies with model size, 3. a family of strong models for text2SQL, 4. a high-quality SFT dataset, 5. a new, robust evaluation metric and kit. They are mostly well-supported by solid experiment results (some of them are known correct to the community as I will mention below).

**Requested Changes:**

1. Currently, I think the related work discussion is insufficient; the empirical study of this work is related to scaling laws and empirical RLVR practices, but only two related papers (in the third paragraph) are discussed. I will suggest the authors to investigate general RLVR empirical studies (some examples can be seen in the reference of the previous section, and Khatri et al. [1]), and position the findings of this paper inside these works.

2. In Fig. 1, the x axis in the model size seems to be in billions. The question is: how does the author know that o3 and GPT-5.2 are 100B models and o4-mini is a 60B model? While people don't know the exact numbers of these proprietary models, some people estimate that these models has trillion-level parameters (https://www.cometapi.com/how-many-parameters-does-gpt-5-have/).

**Minor Changes**

1. Explain in the caption of Tab. 3 on what the ticks and crosses before the models' names represent.

2. Explain why RL 1:1 performs worse than RL 1:4 and RL 1:2 in Fig. 5 w. AVG.

**References**

[1] https://arxiv.org/html/2510.13786v1

---

> ### Author Response · Authors · 2026-03-30
>
> We sincerely thank you for your thorough review and for recognizing the value of our contributions, including our **reward recipe**, the **distilled dataset**, the **evaluation kit**, and our extensive **analysis of RLVR training dynamics**.
>
> We appreciate your constructive feedback and address your specific requests below:
>
> ---
>
> ### **1. Related Work & General RLVR Findings**
>
> We appreciate the pointers to recent literature and will update the Related Work section with the most recent work. While we agree that findings such as RLVR's robustness over SFT are becoming established in the broader community, we view our results as a **strong validation** of these trends within the specific, highly constrained environment of text-to-SQL.
>
> As requested, we will better contextualize our findings, explicitly noting which dynamics are specific to Text-to-SQL and which corroborate known phenomena in the broader RLVR community, adding the following paragraph to the related works:
>
> > *"RLVR in other domains. The efficacy of RLVR has been validated across diverse modalities, mirroring our findings while highlighting Text-to-SQL's unique requirements. Recent studies in Vision-Language-Action (VLA) and VLMs confirm that RL generalizes better than SFT by exploring latent capabilities [1-3]. While research on R1-like models shows that SFT provides an immediate boost [4], other work demonstrates that it can create an "SFT quagmire" in which misleadingly high scores constrain the RL exploration space [6]. Our work confirms this ceiling effect (RQ2), showing that base models often surpass SFT-initialized ones in peak performance. Furthermore, while general RL reasoning may be driven by high-entropy tokens [5], we show that in Text-to-SQL, this is scale-dependent: smaller models require dense, execution-guided feedback to prevent training stalls, whereas larger models thrive on sparse signals and aggressive scaling."*
>
> [1] What Can RL Bring to VLA Generalization? An Empirical Study.
>
> [2] Why Does RL Generalize Better Than SFT? A Data-Centric Perspective on VLM Post-Training.
>
> [3] SFT or RL? An Early Investigation into Training R1-Like Reasoning Large Vision-Language Models.
>
> [4] Good SFT Optimizes for SFT, Better SFT Prepares for Reinforcement Learning.
>
> [5] Beyond the 80/20 Rule: High-Entropy Minority Tokens Drive Effective Reinforcement Learning for LLM Reasoning.
>
> [6] Quagmires in SFT-RL Post-Training: When High SFT Scores Mislead and What to Use Instead.
>
> ---
>
> ### **2. Figure 1 (Proprietary Model Sizes)**
>
> We agree with the reviewer that placing o3, o4-mini, and GPT-5.2 on a specific parameter axis is speculative. Although specified in the caption, we recognize that this approach could be misleading.
>
> To address this, we will **position the proprietary models in a separate, disconnected categorical region** on the right side of the plot to prevent conveying inaccurate parameter estimates, while preserving the visual performance comparison.
>
> ---
>
> ### **3. Table 3 Caption**
>
> We apologize for the missing information. The checkmarks (**✓**) denote models specifically optimized for Text-to-SQL or reasoning via RLVR/specialized training, whereas the crosses (**✗**) denote general-purpose base or instruct models. We will explicitly define this in the caption of Table 3.
>
> ---
>
> ### **Figure 5 (RL 1:1 Performance)**
>
>  As briefly noted in Section 4.4, we hypothesize this is due to stochastic regularization. BIRD-Train has a specific distribution of schemas. Training on the full dataset (1:1) allows the model to overfit to BIRD's specific structures. Subsampling (1:2, 1:4) acts as a regularizer, preventing overfitting and consequently improving zero-shot generalization on out-of-domain datasets such as Spider. To better illustrate our claim, please refer to Table 8 in the appendix.
>
>
> We believe these revisions will make the paper more robust and comprehensive. We thank you again for your help in improving the manuscript.

---

> ### Comment · Reviewer_B7KX · 2026-04-01
>
> Thanks for the authors' response. I remember that authors can update the pdf of their submission; while I think the authors' rebuttal sounds reasonable, I would like to see how the changes to the paper will be applied on the final pdf (prefeably marked in another color from the original text) before stating that my concerns are addressed, since change to the related work are more than one or two sentences. As I do not see the authors' response to other reviewers. while for me the response is "visible to everyone", I assume the authors have not replied to the other reviewers. The authors can do this after discussing with the other reviewers, and I will also check whether their concerns are addressed.

---

> > ### Author Response · Authors · 2026-04-02
> >
> > Thank you for your feedback and for considering our rebuttal reasonable. We would like to address your remaining points:
> >
> > - **Revised Manuscript**: We have uploaded a revised PDF of our submission. All updates, including the expanded Related Work section, are highlighted in blue to distinguish them from the original text.
> >
> > - **Response Visibility**: We will soon post responses to the other reviews, using different colors in the main paper to indicate feedback from each review. We encourage you to review these discussions, as they provide additional context regarding the improvements made to the paper.
> >
> > We look forward to hearing your thoughts on the updated manuscript and are available for further discussion during the reviewer consultation phase.
> >
> > Specifically, the updates to the paper are as follows:
> >
> > - Figure 1 now includes a separate column for unknown model size parameters, indicated as N/A on the X-axis.
> > - The Related Work section now contains a new paragraph titled “RLVR in Other Domains.”
> > - The caption for Table 3 has been updated to include the previously missing information regarding checkmarks.

---

### Review · Reviewer_ovYf · 2026-03-13

**Summary Of Contributions:**

This paper studies how to train Text-to-SQL models with RLVR. I believe the experimental findings of this paper are valuable to the community.

### Strength

- They find that the best reward design depends on model size, where small models need dense, step-by-step rewards to learn effectively, while large models perform better with sparse, pass/fail rewards.

- Cold-Start Insights: Proof that starting RLVR from a base model leads to better final performance than starting from a model already fine-tuned on reasoning traces.

- They release the Think2SQL model family, where their 4B-parameter model performs as well as much larger, closed-source models.

### Weakness
- The study scope of Text-to-SQL is relatively limited.
- Think2SQL-8B underperforms compared to the smaller SQL-R1-7B model (Table 3).

**Audience:**

Yes

**Audience Explanation:**

The topic of RLVR is very popular, and the findings regarding rewards and model size are meaningful.

**Claims And Evidence:**

Yes

**Claims Explanation:**

- The paper clearly shows how model size, reward type, and advantage scaling should be matched for the best results.
- The extensive experiments show that  Think2SQL models' performance and other findings.

**Requested Changes:**

- Explain why Think2SQL-8B slightly underperforms the smaller SQL-R1-7B model. If the main goal is maximizing performance, why not use a more computationally intensive training approach to beat it? The authors are suggested to explain.

- Based on the reward density findings, transitioning from dense to sparse rewards during training seems like a highly relevant and unexplored baseline for a small model. The author can add small experiments or a discussion of this proposal.

- The relationship between dense process rewards and sparse outcome rewards is closely related to previous work [1]. The authors are encouraged to discuss it in the paper.

### Reference
[1] Is PRM Necessary? Problem-Solving RL Implicitly Induces PRM Capability in LLMs. NeurIPS. 2025.

---

> ### Author Response · Authors · 2026-04-08
>
> We thank the reviewer for their insightful feedback and for recognizing the value of our experimental findings on reward design scaling and cold-start dynamics.  All changes are marked in pink in the revised manuscript.
>
> ---
>
> ### 1. Think2SQL-8B vs. SQL-R1-7B
>
> The gap is modest (1.0 pp in weighted average: 69.4 vs. 68.4), while SQL-R1-7B relies on a substantially more data-intensive two-stage pipeline (~200k SFT samples, roughly 40× more supervised data than our setup). Our goal in this work is not exhaustive leaderboard optimization at each scale, but a controlled empirical characterization under comparable budgets. We clarified this trade-off explicitly in Section 4.2.
>
> ---
>
> ### 2. Dense-to-Sparse Reward Curriculum
>
> We thank the reviewer for this insightful suggestion. We conducted the proposed dense-to-sparse curriculum experiments using Qwen3-4B with Group Scaling, training two additional configurations: (i) $R_{GATE}$ → $R_{GATE}$ (dense → dense, as a continuation baseline) and (ii) $R_{GATE}$ → $R_{EXFM}$ (dense → sparse transition). In both cases, training is based on Think2SQL-4B and is trained for one additional epoch using the respective reward. The following table is an extension of Table 2.
>
> | Model | BIRD dev (#1,530) | Spider test (#2,147) | Spider-DK (#1,034) | Spider-Syn (#535) | Spider-Realistic (#508) | EHRSQL dev (#1,008) | Weighted AVG. |
> |---|---|---|---|---|---|---|---|
> | Base | 49.0 (0.5) | 78.6 (0.6) | 68.3 (1.5) | 68.0 (1.7) | 76.5 (0.9) | 29.0 (1.9) | 61.9 |
> | Thinking | 52.0 (0.5) | 80.2 (0.2) | 70.7 (0.5) | 71.4 (1.0) | 78.8 (1.0) | 30.3 (0.8) | 64.1 |
> | $R_{GATE}$ | 59.6 (0.3) | 83.7 (0.2) | **76.0 (0.3)** | **75.0 (0.5)** | 81.2 (0.1) | **33.5 (0.3)** | **68.7** |
> | $R_{GATE}$ + $R_{GATE}$ | 56.4 (0.6) | 81.4 (0.4) | 71.0 (1.2) | 74.2 (0.3) | 79.5 (1.0) | 31.6 (1.3) | 66.1 |
> | $R_{GATE}$ + $R_{EXFM}$ | **60.3 (0.2)** | **83.8 (0.4)** | 74.2 (0.6) | 74.3 (0.6) | **82.1 (0.3)** | 30.3 (0.3) | 68.2 |
>
> The results reveal two findings. First, the dense-to-dense continuation ($R_{GATE}$ → $R_{GATE}$) does not outperform the single-phase dense reward ($R_{GATE}$), underperforming by approximately 2 pp on the weighted average. Second, the dense-to-sparse curriculum ($R_{GATE}$ → $R_{EXFM}$) improves performance relative to the single-phase baseline (59.6 vs. 60.3 on Bird-Dev) but slightly reduces generalization performances.
>
> We note that a more gradual curriculum, e.g., linearly interpolating between dense and sparse rewards, or transitioning at a previous stage of training, may yield different results and represents a promising direction for future work.
>
> To avoid overstating inconclusive results, we discuss this direction explicitly as future work in the revised manuscript.
>
> ---
>
> ### 3. Connection to PRM Literature
> We agree that this line of work is closely related in motivation. Our setting is nonetheless technically different. We do not use a process-reward model or step-level supervision over intermediate reasoning traces. All our rewards are verifiable outcome rewards computed from the final executed SQL output. Thus, we view our findings as complementary rather than overlapping with PRM-style work.
>
> We added a discussion of [1] in Section 2 (Related Work), noting that our finding that dense rewards help small models but hurt large ones may also inform PRM research.
>
> [1] Feng, Zhangying, et al. "Is prm necessary? problem-solving rl implicitly induces prm capability in llms." arXiv preprint arXiv:2505.11227 (2025).
>
> ---
>
> ### 4. Scope of Text-to-SQL
>
> We agree that Text-to-SQL is a focused application domain. Our point is that it is also a practically important and methodologically rich setting for RLVR: it combines structured generation, verifiable execution-based feedback, and challenging cross-schema generalization. We therefore view it as a strong testbed rather than a narrow toy setting. To make this clearer, we revised the introduction to better motivate the practical relevance of Text-to-SQL and to explicitly position our conclusions as a blueprint within this setting. We added this discussion in the Introduction.

---

### Review · Reviewer_oTfm · 2026-03-29

**Summary Of Contributions:**

This paper studies how to make RLVR work better for text-to-SQL, for models under 14B. The main contribution is an empirical blueprint. The authors build a dense execution-based reward from QATCH-style partial-credit signals (cell precision, cell recall, tuple cardinality), plus a gated variant called RGATE. They then run ablations over reward density, advantage normalization, and model size on Qwen3 4B/8B/14B. They also look at cold-start effects, and analyze training-efficiency tradeoffs via subsampling.

The paper's main strength is empirical: it covers reward design, scaling behavior, cold-start effects, and data-efficiency in one coherent study. The practical takeaway (smaller models benefit from denser rewards while larger models tolerate sparser signals) is useful and clearly demonstrated. The evaluation spans multiple benchmarks beyond BIRD, including Spider variants and EHRSQL.

It's also a generally clear and well-written paper!

The weaknesses, roughly in order of severity: the evidence comes from a single model family (Qwen3) which makes the "general blueprint" framing feel overstated; several mechanistic claims are plausible but never directly tested; there are internal inconsistencies in which configuration is reported as optimal for the 14B model; and it's unclear whether the reported variance comes from repeated training runs or only repeated decoding. (More details below.)

**Audience:**

Yes

**Audience Explanation:**

I think it's mostly self-evident: it's about RL and LLMs and adds to the broader empirical knowledge of the community.

**Claims And Evidence:**

Yes

**Claims Explanation:**

The within-family (Qwen3) empirical story is good. The paper clearly shows that at 4B scale dense rewards outperform sparse EX-based rewards, and that RLVR improves over the base/thinking variants. The cold-start section supports the narrower claim that for Qwen3-4B, starting RLVR from the base model can outperform starting from SFT or "thinking" initializations. The subsampling experiment supports the observation that moderate subsampling can preserve or even improve generalization in this setup.

The paper often presents causal explanations where the evidence is only correlational. The claims about zero-variance groups, exploration penalties, convergence traps, and dense-floor reward hacking are sensible hypotheses, but the paper doesn't directly measure any of them. I'd want to see at least one concrete analysis per claim before accepting these as explanations rather than speculation. Or just cut them?

The Pareto section says RLVR "consistently outperforms SFT across the Pareto frontier," but Figure 5 includes at least one RL regime that's worse than SFT, so that claim needs qualifying.

The text says the final Think2SQL models for 8B and 14B use sparse reward with Batch Scaling, but the reported Think2SQL-14B row looks like it lines up more closely with a Group Scaling configuration, and Table 6 suggests the 14B story isn't as clean as the prose implies. That doesn't invalidate the paper's main intuition, but it does reduce my confidence in the headline numbers.

The comparisons to proprietary frontier models are interesting but inherently prompt-sensitive and single-pass. I'd treat them as suggestive context rather than decisive evidence of parity. I know you quoted "cost considerations" - how much would it actually cost to go more deeply here? Is it thousands of dollars or perhaps a few hundred?

**Requested Changes:**

Five changes are critical:

(1) Resolve the Think2SQL-14B configuration inconsistency. State the exact model-selection rule and correct whichever of the prose or tables is wrong.

(2) Clarify what the reported standard deviations represent: three independent training runs or three inference runs with a fixed checkpoint.

(3) Either add diagnostics (entropy curves, group-variance histograms, reward-distribution plots) that support the causal claims, or rewrite them as hypotheses.

(4) Qualify the Pareto-dominance claim to account for the RL regimes in Figure 5 that underperform SFT.

(5) Narrow the headline framing to Qwen3, or add at least one non-Qwen backbone to the main experiments. The limitation is acknowledged in Section 6 but the title and abstract still read as model-general.

Non-critical changes that would strengthen the work:

(1) Report standard EX alongside refined EX in the main body, or justify why refined EX should be the primary metric.

(2) Give more detail on baseline evaluation fairness, especially for proprietary models and reproduced baselines that look unexpectedly weak. A short prompt-sensitivity or decoding-sensitivity analysis would help.

(3) Add qualitative examples showing when the dense reward helps training and when it misleads the model despite low exact execution correctness.

(4) Make the efficiency discussion more apples-to-apples by clarifying which costs are counted and which are excluded when comparing against SFT-heavy baselines.

(5) Add error bars to figures so the visual takeaways are clearer.

---

> ### Author Response · Authors · 2026-04-08
>
> Thank you for your meticulous and constructive review. All changes are marked in orange in the revised manuscript.
>
> ---
>
> ### Causal claims (Critical #3).
> We softened language in Sections 4.2–4.3 and added three diagnostics in Appendix F:
> - Figure 10 to study the zero variance groups across reward formulations (fraction of zero-reward groups higher for sparse rewards),
> - Figure 11 to detect any exploration penalty (entropy and reward curves across cold-start configurations showing entropy diminishing for SFT/Thinking but steady increase for base),
> - and Figure 12 to detect possible dense-floor reward hacking  (gradient norms and completion lengths showing $R_{GATE}$ causes persistent gradient spikes in 14B but smooth convergence in 4B).
>
> These ground the previously speculative mechanisms as empirically supported hypotheses.
>
> ---
>
> ### Pareto-dominance (Critical #4).
> The RQ3 summary in Section 4.4 now states: "While RLVR mostly outperforms SFT across the Pareto frontier, extreme subsampling can lead to underperformance relative to SFT", reflecting $RL_{1:16}$ (52.2) falling below $SFT_{1:1}$ (55.5) in Figure 5.
>
> ---
>
> ### Think2SQL-14B inconsistency (Critical #1).
> The correct configuration is $R_{EXFM}$ with Group Scaling (not Batch Scaling). Corrected in Section 4.2. The confusion arose because Batch Scaling achieves a comparable BIRD-Dev score, but Group Scaling has a higher OOD average (74.0 vs. 71.9). We added Weighted Average and OOD Average columns to Table 6 and now state the selection rule explicitly: peak BIRD-Dev, with OOD average as tiebreaker within one standard deviation.
>
> ---
>
> ### Proprietary comparisons (Critical #5 context).
> We report total API cost (~$300) in Section 4.1 and add a limitations paragraph in Section 6 acknowledging that the unified prompt strategy may disadvantage some models.
>
> ---
>
> ### Headline framing (Critical #5).
> The abstract now explicitly scopes claims to "the Qwen3 model family." Section 6 notes that the strategies are architecturally agnostic in principle, but our evidence is limited to Qwen3.
>
> ---
>
> ### Standard vs. Refined EX (Non-critical #1).
> Section 4.1 now includes two concrete examples of situations in which standard EX fails. Table 6 (Appendix A) reports standard EX results, confirming that relative rankings are preserved.
>
> ---
>
> ### Evaluation fairness (Non-critical #2).
> Specialized models utilize their original prompts, while general-purpose models adopt the standardized prompt [1]. We recognize that prompt sensitivity may affect performance, but systematic prompt selection is beyond the scope of this paper. Consequently, we report only the standardized prompt from the literature [1].
>
> [1] Haoyang Li et al. 2025. OmniSQL: Synthesizing High-Quality Text-to-SQL Data at Scale. Proc. VLDB Endow. 18, 11 (July 2025)
>
>
> ---
>
> ### Standard deviations (Critical #2).
> Section 4.1 now explicitly states that all reported std values are obtained from repeated stochastic decoding of a fixed checkpoint (n=3), rather than from independent training reruns.
>
> ---
>
> ### Efficiency comparison (Non-critical #4).
> Section 4.4 now specifies that wall-clock times cover reward computation (CPU), vLLM rollout generation (4×H100), and policy updates (4×H100) for RLVR, and forward/backward passes (4×H100) for SFT. Data preprocessing, distillation, and evaluation costs are excluded from both.
>
> ---
>
> ### Error bars for figures (Non-critical #5):
> We thank the reviewer for this suggestion. We report mean ± std across three runs for all configurations in the corresponding tables (Tables 2, 3, 4, 6, 7, 8), ensuring that full numerical uncertainty is accessible. We opted against overlaying error bars on the summary figures (e.g., Figures 3 and 5) because the standard deviations are consistently small (typically ≤ 1 pp), rendering error bars largely imperceptible at the figure scale. We believe the current combination of detailed tabular uncertainty and clean visual summaries offers the most readable presentation.
>
> ---
>
> ### Add qualitative examples (Non-critical #3):
> Appendix C now reports two reasoning examples: one that erroneously causes the model to fail (Fig 9) and one that correctly points to the solution (Fig 8).

---

> > ### Comment · Reviewer_oTfm · 2026-04-08
> >
> > Thank you for your thorough reply.

---

### Author Response · Authors · 2026-04-08

We thank the reviewers for their constructive feedback and for helping improve this paper. The following summarizes the strengths highlighted by the reviewers and the key updates made to the paper.

The reviewers highlight several strengths: (i) the paper is well-written and easy to follow (B7KX, oTfm); (ii) the empirical study is solid and comprehensive, covering reward design, scaling behavior, cold-start effects, and data efficiency in one coherent study (oTfm, B7KX); (iii) the practical finding that smaller models benefit from denser rewards while larger models tolerate sparser signals is useful and well demonstrated (ovYf); (iv) the cold-start insight that base-model initialization outperforms SFT/thinking initialization is valuable (ovYf); (v) and the paper delivers multiple reusable contributions, a reward recipe, a model family, a high-quality dataset, and an evaluation toolkit (B7KX).

The following summary outlines the key updates made to the paper:
- Resolved the Think2SQL-14B configuration inconsistency and stated the model-selection rule explicitly (oTfm #1)
- Clarified that reported standard deviations come from repeated stochastic decoding, not independent training runs (oTfm #2)
- Added three diagnostic figures (Appendix F), grounding previously speculative causal claims as empirical hypotheses (oTfm #3)
- Qualified the Pareto-dominance claim to account for RL regimes that underperform SFT (oTfm #4)
- Scoped headline framing to the Qwen3 model family in the abstract and introduction (oTfm #5)
- Added the rationale for refined EX in section 4.1 (oTfm NC#1)
- Documented evaluation fairness considerations and API costs for proprietary comparisons (oTfm NC#2, NC#4)
- Added error std values in tables (oTfm NC#5)
- Explained the Think2SQL-8B vs. SQL-R1-7B gap in terms of data budget trade-offs (ovYf #1)
- Added discussion of the PRM literature connection in Related Work (ovYf #3)
- Expanded Related Work with recent general RLVR empirical studies to better position our findings (B7KX #1)
- Revised Figure 1 to avoid speculative parameter estimates for proprietary models (B7KX #2)
- Clarified Table 3 caption symbols and the $RL_{1:1}$ vs. $RL_{1:2/1:4}$ generalization dynamic (B7KX minor)

---

### Decision · Action_Editor_4F8T · 2026-04-30

**Recommendation:** Accept as is

**Audience:**

Yes

**Audience Explanation:**

The paper examines the reasoning capabilities of large language models in complex, multi-table environments. It spans multiple topics, such as post-training, Text2SQL and reasoning, and is therefore relevant to several subgroups of the TMLR audience.

**Claims And Evidence:**

Yes

**Claims Explanation:**

The paper supports its central claims, for example, that the proposed dense reward function improves upon the sparse one, through empirical evidence using the Qwen3 model family. Reviewers’ concerns about the empirical evaluation were addressed during the rebuttal.